# Making Classic GNNs Strong Baselines Across Varying Homophily: A Smoothness–Generalization Perspective

**Ming Gu[★♠], Zhuonan Zheng[★♠], Sheng Zhou[★‡], Meihan Liu[†],**
**Jiawei Chen[♠], Qiaoyu Tan[§], Liangcheng Li[★♠,∗], Jiajun Bu[★♠]**
[★] Zhejiang Key Laboratory of Accessible Perception and Intelligent Systems, Zhejiang University
[♠]College of Computer Science and Technology, Zhejiang University
[‡]School of Software Technology, Zhejiang University
[†]China University of Mining and Technology
[§] Department of Computer Science, New York University Shanghai

## Abstract

Graph Neural Networks (GNNs) have achieved great success but are often considered to be challenged by varying levels of homophily in graphs. Recent *empirical* studies have surprisingly shown that homophilic GNNs can perform well across datasets of different homophily levels with proper hyperparameter tuning, but the underlying theory and effective architectures remain unclear. To advance GNN universality across varying homophily, we theoretically revisit GNN message passing and uncover a novel *smoothness-generalization dilemma*, where increasing hops inevitably enhances smoothness at the cost of generalization. This dilemma hinders learning in high-order homophilic neighborhoods and all heterophilic ones, where generalization is critical due to complex neighborhood class distributions that are sensitive to shifts induced by noise or sparsity. To address this, we introduce the Inceptive Graph Neural Network (IGNN) built on three simple yet effective design principles, which alleviate the dilemma by enabling distinct hop-wise generalization alongside improved overall generalization with adaptive smoothness. Benchmarking against 30 baselines demonstrates IGNN's superiority and reveals notable universality in certain homophilic GNN variants. Our code and datasets are available at https://github.com/galogm/IGNN.

## 1 Introduction

Graph Neural Networks (GNNs) [1–4] have attracted substantial attention, achieving notable success across various domains [5–8]. Broadly, GNNs are classified into homophilic GNNs (homoGNNs) [9] and heterophilic GNNs (heteroGNNs) [10]. HomoGNNs operate under the homophily assumption, which posits that adjacent nodes tend to share similar labels. In contrast, heteroGNNs are tailored for heterophilic graphs, where connected nodes are more likely to have differing labels.

However, real-world graphs do not exhibit a clear dichotomy between homophily and heterophily, but instead present a continuous spectrum. As illustrated in Figure 1a and 1b, ***varying homophily appears within a single graph across hops and nodes***. Therefore, it is essential to develop GNNs that generalize to different levels of homophily, rather than making separate designs for homophily and heterophily as in existing methods. Recent studies [11] have *empirically* shown that homoGNNs, after hyperparameter tuning with residual connections and dropout, can outperform advanced methods designed for heterophily. This suggests that homoGNNs possess an inherent potential to adapt to

---

[∗]Corresponding Author: liangcheng_li@zju.edu.cn.

39th Conference on Neural Information Processing Systems (NeurIPS 2025).

varying homophily, but the underlying theory and effective architectures remain unclear. A question arises: *What enables universality across varying homophily in GNNs, or even in homoGNNs?*

To gain a deeper understanding, we theoretically revisit the classic GNN message-passing process and identify a novel ***smoothness-generalization dilemma***, as depicted in Figure 1c. Here, *smoothness* refers to the alignment of node representations within neighborhoods, while *generalization* denotes the ability to handle distribution shifts across neighborhoods. *As the number of hops increases, smoothness inevitably rises, while generalization correspondingly declines due to the intrinsic trade-off between the two.* This dilemma is negligible in low-order homophilic neighborhoods, where strong homophily naturally aligns with smoothness, rendering generalization less critical. However, it becomes detrimental in higher-order homophilic neighborhoods and all heterophilic ones. We show that strong generalization is crucial in these cases to address complex neighborhood class distributions, which are highly sensi-

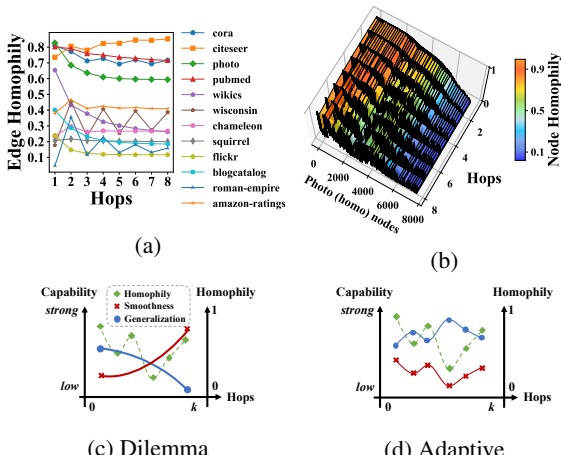

Figure 1: Varying homophily across (a) hops or (b) nodes. Conceptual illustration of the theoretical insight: (c) Smoothness-Generalization dilemma identified in GNNs; (d) Expected adaptive capabilities for varying homophily.

tive to shifts induced by noise or sparsity. Yet, it remains constrained by the increasing smoothness imposed by the dilemma. *This insight suggests that resolving the smoothness-generalization dilemma can benefit both homophilic and heterophilic settings without requiring separate designs (Figure 1d), thereby unlocking the full potential of classic GNNs and paving the way toward achieving universality.*

"*More is in vain when less will serve, for Nature is pleased with simplicity*" [12], echoing Sir Isaac Newton, we seek to *make **minimal changes*** to classic GNNs to reveal the dilemma as a fundamental impediment to universality. We introduce Inceptive Graph Neural Network (IGNN), where the term *inceptive* [13] signifies concurrent learning of multiple receptive fields. IGNN is built upon three minimal design principles: separative neighborhood transformation (SN), inceptive neighborhood aggregation (IN), and neighborhood relationship learning (NR). Theoretically and empirically, we demonstrate that these changes alleviate the dilemma from two perspectives: ***First***, *inceptive neighborhood relationship learning, IN &NR, enable GNNs to approximate arbitrary graph filters for adaptive smoothness capabilities.* ***Second***, *incorporating SN allows distinct hop-wise generalization and improved overall generalization.* Our main contributions are:

- **Theoretical Insights**. We advance the theoretical understanding of GNN universality across varying levels of homophily by uncovering the smoothness-generalization dilemma, providing a foundation for theoretically grounded universal designs.
- **Universal Framework**. We introduce IGNN, a universal message-passing framework based on three minimal yet effective design principles. IGNN mitigates the dilemma without relying on specialized modules tailored for either homophilic or heterophilic graphs.
- **Benchmark and Empirical Findings**. We establish a comprehensive benchmark consisting of 30 representative baselines to assess the effectiveness of our design principles. Our results demonstrate that not only can classic GCNs enhanced with these principles achieve state-of-the-art (SOTA) performance, but also that certain existing homoGNNs inherently possess universal capabilities.

## 2 Related Works

**Homophilic Graph Neural Networks**. GNNs have demonstrated remarkable abilities in managing graph-structured data, particularly under the assumption of homophily. Traditional GNNs can be broadly categorized into two categories. Spectral GNNs, such as the GCN [2], leverage various graph filters to derive node representations. In contrast, spatial GNNs aggregate information from

neighboring nodes and combine it with the ego node to update representations, employing methods such as attention mechanisms [3] and sampling strategies [9]. Unified frameworks [14, 15] have been proposed to integrate and elucidate these diverse message-passing approaches. Several multi-hop techniques were proposed to address the limitations of long-range dependencies, such as residual connections [16] and jumping knowledge [17]. *However, these homophilic methods are often considered less effective when dealing with heterophilic settings, while a recent empirical study shows its potential to universality [11] but lacks a theoretical understanding.*

**Heterophilic Graph Neural Networks**. Addressing the challenges posed by heterophily, several innovative approaches have been proposed: (1) Neighborhood extension: Techniques such as high-order neighborhood concatenation [10, 18], neighborhood discovery [19], neighborhood refinement [20], and global information capture [21]. (2) Neighborhood discrimination: Methods including ordered neighborhood encoding [22], ego-neighbor separation [10], and hetero-/homo-phily neighborhood separation [23]. (3) Fine-grained information utilization: Strategies such as multi-filter signal usage [24, 25], intermediate layer combination [10], and refined gating or attention mechanisms [26]. These methods generally retain the practice of message passing [27] that aggregates multi-hop neighborhood information. *However, these methods often treat homophily and heterophily separately, leading to a paradox: effectively separating them would require prior knowledge of node labels, while it is precisely the labels that need to be learned.* A holistic understanding is needed to guide the development of an architecture that adapts to both settings without different treatments.

**Oversmoothing, Heterophily and Generalization**. Early studies [28–30] investigate oversmoothing or generalization without considering varying homophily, while later works reveal that oversmoothing and heterophily are often intertwined leaving generalization unexamined. Bodnar et al. [31] attribute both oversmoothing and heterophily to the underlying graph geometry using a sheaf-based formulation. Park et al. [32] counter the two by reversing the diffusion process, yet their approach remains architecturally motivated without theoretical insight into generalization. Meanwhile, several heterophily-oriented models [22, 25, 33] have been shown to alleviate oversmoothing, while oversmoothing-focused designs [16, 34] also perform well under heterophily. In contrast, Ma et al. [35] explore the link between heterophily and generalization while omitting oversmoothing. In summary, *existing studies have examined all pairwise combinations among oversmoothing, heterophily, and generalization, yet no unified framework has bridged all three.* We fill this gap through a unified theoretical lens, demonstrating that the issues of oversmoothing, poor generalization, and heterophily all stem from a shared underlying trade-off between smoothness and generalization, thereby offering a principled foundation for a unified understanding and guides the design of more universal GNNs.

## 3 Notations and Preliminaries

Given an undirected graph $\mathcal{G}(\mathcal{V}, \mathbf{X}, \mathcal{E}, \mathbf{A})$ with the node set $\mathcal{V} = \{v_1, \ldots, v_N\}$ and feature matrix $\mathbf{X} = [\mathbf{x}_0, \ldots, \mathbf{x}_N]^\top \in \mathbb{R}^{N \times D}$, the edge set $\mathcal{E}$ is represented by the adjacency matrix $\mathbf{A} \in \mathbb{R}^{N \times N}$. $\mathbf{A}_{ij} = 1$ if $(v_i, v_j) \in \mathcal{E}$, otherwise $\mathbf{A}_{ij} = 0$. The degree matrix is $\mathbf{D} = \text{diag}(d_1, \ldots, d_N) \in \mathbb{R}^{N \times N}$, $d_i = \sum_j^N \mathbf{A}_{ij}$. The re-normalization of $\mathbf{A}$ is $\widehat{\mathbf{A}} = \widehat{\mathbf{D}}^{-\frac{1}{2}} (\mathbf{A} + \mathbf{I}_N) \widehat{\mathbf{D}}^{-\frac{1}{2}}$, where $\mathbf{I}_N$ is the identity matrix. The symmetrically normalized graph Laplacian matrix is $\widehat{\mathbf{L}} = \mathbf{I}_N - \widehat{\mathbf{A}}$. Edge and node homophily are computed as: $h_e = (1/|\mathcal{E}|) \sum_{(v_i, v_j) \in \mathcal{E}} \mathbb{I}(c_i = c_j)$, $h_n = 1/N \sum_{v_i \in \mathcal{V}} \sum_{(v_i, v_j) \in \mathcal{E}} \mathbb{I}(c_i = c_j)/d_i$.

### 3.1 Smoothness of GNNs

Oono and Suzuki [29] describe the smoothness characteristic of GNNs with information loss from $\mathbf{X}$ on asymptotic behaviors of GNNs from a dynamical systems perspective. They demonstrate that when it extends with more layers, the GNN representation (i.e., $\mathbf{H}_G^{(k)} = \sigma(\widehat{\mathbf{A}} \mathbf{H}^{(k-1)} \mathbf{W}^{(k)})$, see Section 4) exponentially approaches information-less states, which is a subspace $\mathcal{M}$ in Definition 3.1.

**Definition 3.1** (subspace). Let $\mathcal{M} := \left\{ \mathbf{E} \mathbf{B} \mid \mathbf{B} \in \mathbb{R}^{M \times D} \right\}$ be an $M$-dimensional subspace in $\mathbb{R}^{N \times D}$, where $\mathbf{E} \in \mathbb{R}^{N \times M}$ is orthogonal, i.e. $\mathbf{E}^\mathrm{T} \mathbf{E} = \mathbf{I}_M$, and $M \leq N$.

Following their notations, we denote the maximum singular value of $\mathbf{W}^{(l)}$ by $s_l$ and set $s := \sup_{l \in \mathbb{N}_+} s_l$. Denote the distance that induced as the Frobenius norm from $\mathbf{X}$ to $\mathcal{M}$ by $d_{\mathcal{M}}(\mathbf{X}) := \inf_{\mathbf{Y} \in \mathcal{M}} \|\mathbf{X} - \mathbf{Y}\|_\mathrm{F} = \mathcal{D}$. The following Corollary 3.2 shows the information loss as layer $l$ goes.

**Corollary 3.2** (Oono and Suzuki [29]). *Let $\lambda_1 \leq \cdots \leq \lambda_N$ be the eigenvalues of $\widehat{A}$, sorted in ascending order. Suppose the multiplicity of the largest eigenvalue $\lambda_N$ is $M(\leq N)$, i.e., $\lambda_{N-M} < \lambda_{N-M+1} = \cdots = \lambda_N$ and the second largest eigenvalue is defined as $\lambda := \max_{n=1}^{N-M} |\lambda_n| < |\lambda_N| = 1$. Let $\mathbf{E}$ to be the eigenspace associated with $\lambda_{N-M+1}, \cdots, \lambda_N$. Then we have $\lambda < \lambda_N = 1$, and*

$$d_{\mathcal{M}}\left(\mathbf{H}^{(l)}\right) \leq s_l \lambda d_{\mathcal{M}}\left(\mathbf{H}^{(l-1)}\right), \tag{1}$$

*where $\mathcal{M} := \left\{ \mathbf{EB} \mid \mathbf{B} \in \mathbb{R}^{M \times D} \right\}$. If $s_l \lambda < 1$, the l-th layer output exponentially approaches $\mathcal{M}$.*

***Greater smoothness with larger information loss is indicated by a smaller distance $d_{\mathcal{M}}(\mathbf{H}^{(l)})$ from the representations to the subspace $\mathcal{M}$** [29]. This is because the subspace denotes the convergence state of minimal information retained from the original node features $\mathbf{X}$, with the only information of the connected components and node degrees of $\widehat{\mathbf{A}}$. This means that for any $\mathbf{Y} \in \mathcal{M}$, if two nodes $v_i, v_j \in \mathcal{V}$ are in the same connected component and their degrees are identical, then the corresponding column vectors of $\mathbf{Y}$ are identical, i.e., they cannot be distinguished.*

### 3.2 Generalization of GNNs

GNN generalization can be governed by the Lipschitz constant as discussed in existing works [36, 37]:

**Definition 3.3** (Lipschitz constant). A function $f : \mathbb{R}^n \to \mathbb{R}^m$ is called Lipschitz continuous if there exists a constant $L$ such that $\forall x, y \in \mathbb{R}^n, \|f(x) - f(y)\|_2 \leq L\|x - y\|_2$, where the smallest $L$ for which the previous inequality is true is called the Lipschitz constant of $f$ and denoted $\hat{L}$.

***Better generalization is exhibited by GNNs with a smaller Lipschitz constant $\hat{L}$** [38]. This paper does not discuss generalization on graph domain adaption [39], but discusses generalization regarding inherent structural disparity [40] and data distribution shifts from training to test sets [38].*

## 4 Theoretical Analysis of Classic GNNs

Generally, most GNNs capture multi-hop information by stacking message-passing (MP) layers [41]:

$$\mathbf{h}_v^{(0)} = \mathbf{x}_v, \ \mathbf{m}_v^{(k)} = \text{AGG}^{(k)}(\{\mathbf{h}_u^{(k-1)} \mid u \in \mathcal{N}(v)\}), \ \mathbf{h}_v^{(k)} = \text{COM}^{(k)}(h_v^{(k-1)}, m_v^{(k)}), \tag{2}$$

where $\mathbf{h}_v^{(k)}$ is the hidden representation and $\mathbf{m}_v^{(k)}$ is the message for node $v$ in the $k$-th layer. $\text{AGG}(\cdot)$ and $\text{COM}(\cdot)$ denote the aggregation and combination function, while $\mathcal{N}(v)$ is the set of neighbors adjacent to node $v$. Denoting $\mathbf{H}^{(k)} = [\mathbf{h}_0^{(k)}, \mathbf{h}_1^{(k)}, \cdots, \mathbf{h}_N^{(k)}]^\top \in \mathbb{R}^{N \times F}$, the widely used GCN implementation can be written as $\mathbf{H}_G^{(k)} = \sigma(\widehat{\mathbf{A}}\mathbf{H}^{(k-1)}\mathbf{W}^{(k)})$, where $\sigma(\cdot)$ is the activation function.

### 4.1 Smoothness-Generalization Dilemma

The following Theorem 4.1 reveals a dilemma in classic GCNs of $k$ layers. See proof in Appendix A.1.

**Theorem 4.1.** *Given a graph $\mathcal{G}(\mathbf{X}, \mathbf{A})$, let the representation obtained via $k$ rounds of GCN message passing on symmetrically normalized $\widehat{\mathbf{A}}$ be denoted as $\mathbf{H}_G^{(k)} = \sigma(\widehat{\mathbf{A}}\mathbf{H}^{(k-1)}\mathbf{W}^{(k)})$, and the Lipschitz constant of this $k$-layer graph neural network be denoted as $\hat{L}_G$. Given the distance from $\mathbf{X}$ to the subspace $\mathcal{M}$ as $d_{\mathcal{M}}(\mathbf{X}) = \mathcal{D}$, then the distance from $\mathbf{H}_G^{(k)}$ to $\mathcal{M}$ satisfies:*

$$d_{\mathcal{M}}(\mathbf{H}_G^{(k)}) \leq \hat{L}_G \lambda^k \mathcal{D}, \tag{3}$$

*where $\hat{L}_G = \|\prod_{i=0}^k \mathbf{W}^{(i)}\|_2$, and $\lambda < 1$ is the second largest eigenvalue of $\widehat{\mathbf{A}}$.*

**Corollary 4.2.** *$\forall \hat{L}_G, \epsilon > 0, \exists k^* = \lceil (\log \frac{\epsilon}{\hat{L}_G \mathcal{D}})/\log \lambda \rceil$, such that $d_{\mathcal{M}}(\mathbf{H}_G^{(k^*)}) < \epsilon$, where $\lceil \cdot \rceil$ is the ceil of the input.*

**Remark**. As $\mathcal{D}$ is constant with respect to $\mathbf{X}$, we observe that the distance is upper-bounded by three factors: the second largest eigenvalue $\lambda$ of $\widehat{\mathbf{A}}$, the Lipschitz constant $\hat{L}_G$ corresponding to the norm of the product of all $\mathbf{W}^{(i)}$, and the layer depth $k$. Several conclusions can be drawn.

***First, there exists a smoothness-generalization dilemma***. Since $\lim_{k\to\infty}\lambda^k = 0$, $\hat{L}_G$ has to rise when $k$ increases to prevent $d_{\mathcal{M}}(\mathbf{H}_G^{(k)})$ from convergent to 0. This is evidenced by the upper bound of the Lipschitz constant continuing to increase as training progresses [37]. However, a large $\hat{L}_G$ implies reduced generalization, leading to a significant performance gap between training and test accuracy [38]. Consequently, either oversmoothing or poor generalization will occur at large $k$.

***Second***, from Corollary 4.2, we see that for any given $\hat{L}_G$, there exists a $k$ such that the distance from the representations to the subspace is smaller than any arbitrarily small $\epsilon$. Thus, extremely small distance with indistinguishable representations becomes inevitable for sufficiently large $k$, as $\hat{L}_G$ computing from weight matrices can not be infinitely large due to the finite computational precision.

In summary, although oversmoothing has been associated with generalization before [29], this dilemma reveals a more intricate balance in an *either-or* situation. When the classic GCN attempts to counter oversmoothing and recover discriminative representations from the over-smoothed $\mathbf{A}^k\mathbf{X}$ by increasing the spectral norm of $\mathbf{W}^{(i)}$, the resulting larger Lipschitz constant inevitably worsens generalization. Conversely, constraining the norm of $\mathbf{W}^{(i)}$ to maintain a low Lipschitz constant and preserve generalization prevents the model from effectively reversing the over-smoothed $\mathbf{A}^k\mathbf{X}$, yielding indistinguishable node embeddings. *This interplay constitutes* **the core of the smoothness–generalization dilemma**: *efforts to improve one aspect inherently compromise the other*.

### 4.2 How this Dilemma Hinders Performance across Varying Homophily

Next, we bridge the smoothness-generalization dilmma with varying homophily to elucidate *the intrinsic relationship among oversmoothing, generalization, and heterophily*. In essence, graph learning requires adaptive capabilities in both smoothness and generalization for neighborhoods of varying homophily. Table 1 summarizes these dilemma impacts.

*In homophilic settings, the dilemma primarily affects high-order neighborhoods, whereas low-order ones are less impacted.* This can be intuitively understood as smoothness and generalization aligning in low-order homophilic neighborhoods, which always favors pulling together the representations of same-label nodes within these hops. However, smoothness begins to conflict with generalization in

Table 1: Dilemma Impacts. S. and G. are short for smoothness and generalization, while **+**, **−** and $\sim$ denote strong, poor and adaptive capability. ○ means inconsequential (when S. aligns with the homophily bias).

| Homophily | Oversmoothing | Low Orders | | | High Orders | | |
|---|---|---|---|---|---|---|---|
| Heterophily | Mixed | $h_e$ | S. | G. | $h_e$ | S. | G. |
| **Classic MP Capability** | | | + | − | | + | − |
| Learning | Homo | high | + | ○ | low/varying | − / ∼ | + |
| Requirements | Hetero | low/varying | − / ∼ | + | low/varying | − / ∼ | + |

high orders of low or varying homophily, as bringing closer nodes of different labels in these neighborhoods is detrimental. This discrepancy in generalization is clearly exemplified in PMLP [42].

*In heterophilic settings, the dilemma exhibits negative effects across both low- and high-order neighborhoods.* **First**, the complex neighborhood class distribution (NCD) [35] in heterophilic neighborhoods makes it easy for noise or even sparsity to result in mismatched or incomplete NCDs for nodes of the same label, which requires strong generalization ability to mitigate. A toy example in Figure 2 demonstrates that heterophilic neighborhoods suffer from larger NCD shifts caused by the same sparsity, as evidenced by larger distribution variances $s^2_{hetero}$ both in hop 1 and 2 compared to those $s^2_{homo}$ of homophilic neighborhoods. **Second**, there is a greater structural inconsistency between the training

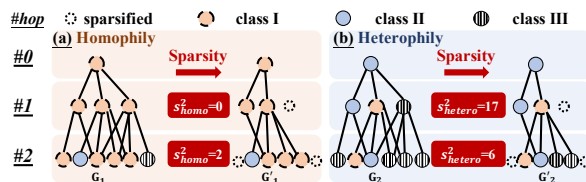

Figure 2: Toy Example of the Sparsity Influence. Three nodes at the *same positions* are sparsified from the (a) homo- and (b) hetero-philic neighborhoods of the *same structure*. Statistics of the neighborhood information and NCD shift variances $s^2$ are presented as:

| | Neigbors | $G_1$ NCD | Neigbors | $G_1'$ NCD | Neigbors | $G_2$ NCD | Neigbors | $G_2'$ NCD |
|---|---|---|---|---|---|---|---|---|
| class | I II III | [I, II, III] | I II III | [I, II, III] | I II III | [I, II, III] | I II III | [I, II, III] |
| hop 1 | 3 0 0 | [1,0,0] | 2 0 0 | [1,0,0] | 1 1 1 | $[\frac{1}{3},\frac{1}{3},\frac{1}{3}]$ | 1 1 0 | $[\frac{1}{2},\frac{1}{2},0]$ |
| hop 2 | 8 1 1 | [0.8,0.1,0.1] | 4 1 0 | [0.8,0.2,0] | 1 3 6 | [0.1,0.3,0.6] | 1 2 2 | [0.2,0.4,0.4] |
| hop 1 | $s^2_{homo} = \|[1,0,0] - [1,0,0]\|^2 * 100 = 0$ | | | | $s^2_{hetero} = \|[\frac{1}{3},\frac{1}{3},\frac{1}{3}] - [\frac{1}{2},\frac{1}{2},0]\|^2 * 100 = 17$ | | | |
| hop 2 | $s^2_{homo} = \|[0.8,0.1,0.1] - [0.8,0.2,0]\|^2 * 100 = 2$ | | | | $s^2_{hetero} = \|[0.1,0.3,0.6] - [0.2,0.4,0.4]\|^2 * 100 = 6$ | | | |

and test sets in heterophilic graphs compared to homophilic ones [40], as heterophilic graphs exhibit a mixture of homophilic and heterophilic patterns, which also requires good generalization.

In summary, *the **core insight** is that challenges are posed by the smoothness-generalization dilemma in both homophily and heterophily, resulting in the absence of universality across varying homophily.*

## 5 Making Classic GNNs Strong Baselines: Inceptive Message Passing

An ***intuitive approach*** to addressing the dilemma is to (1) decouple *smoothness* and *generalization* from a rigid trade-off, endowing them with the capacity to adapt independently to varying homophily; and (2) preserve the embeddings of low/medium orders, acknowledging that oversmoothing is inevitable at sufficiently large hops. To this end, we propose a unified message-passing architecture termed I̲nceptive G̲raph N̲eural N̲etworks (**IGNN**), which is designed to realize this adaptivity with ***minimal cost***. Instead of introducing additional complex modules, IGNN can *easily empower even the classic GNNs by addressing the dilemma through **three simple yet effective design principles***.

### 5.1 Inceptive GNN Framework (IGNN)

**Separative Neighborhood Transformation (SN)** avoids sharing or coupling transformation layers across neighborhoods: $\mathbf{h}_v^{(k)} = f^{(k)}(\mathbf{x}_v) = \mathbf{x}_v \mathbf{W}^{(k)}$, where $f^{(k)}(\cdot)$ is the transformation for the $k$-th neighborhood. The absence of SN implies all $k$-hop neighborhood transformations either share the same parameters $\mathbf{W}_\theta$ or are cascade-coupled in a multiplicative manner, such as $\prod_i^k \mathbf{W}^i$ (see Appendix D.1). This design aims to capture the unique characteristics of each neighborhood, enabling personalized generalization capability with distinct Lipschitz constants for each neighborhood.

**Inceptive Neighborhood Aggregation (IN)** simultaneously embeds different receptive fields: $\mathbf{m}_v^{(k)} = \mathbf{AGG}^{(k)}\left(\{\mathbf{h}_u^{(k)} \mid u \in \mathcal{N}_v^{(k)}\}\right)$, where $\mathbf{AGG}^{(k)}(\cdot)$ represents the neighborhood aggregation function of the $k$-th hop. The simplest approach involves partitioning the $k$-th order rooted tree of neighborhoods into $k$ distinct neighborhoods $\mathcal{N}_v^{(k)} = \mathcal{N}_v(\mathbf{A}^k)$ with $\mathcal{N}_v^{(0)} = \{v\}$. The inceptive nature of the architecture preserves the embedding of low orders and prevents high-order neighborhood representations from being computed based on low-order ones, which avoids cascading the learning of different hops and propagating errors if one becomes corrupted. Moreover, it prevents the product-type amplification of the Lipschitz constant (Theorem 4.1 and 5.3), which would otherwise limit the generalization ability. Notably, some *dynamic message-passing methods* [18, 43] unconstrained by the fixed neighborhood structure $\mathbf{A}$ can be viewed as advanced variants of inceptive architectures with *skip connections* [17, 44]. However, as our goal is to enhance classical GNNs with minimal overhead rather than adopt complex dynamic aggregations, we do not employ them in IGNN.

**Neighborhood Relationship Learning (NR)** adds a neighborhood-wise relationship learning module to learn the correlations among neighborhoods: $\mathbf{h}_v = \mathbf{REL}\left(\{\mathbf{m}_v^{(k)} \mid 0 \leq k \leq K\}\right)$, where $\mathbf{REL}(\cdot)$ is the r̲elationship learning function of multiple neighborhoods. The relationships among various neighborhoods represent a new characteristic in IGNN, extending the combination field from a single neighborhood of ego and neighboring nodes in COM$(\cdot)$ to multiple neighborhoods of various hops in REL$(\cdot)$. Based on the learning mechanism of relationships, IGNN can be divided into three variants.

A brief overview of the variants is presented in Table 2 with a comparison in Appendix D.1. ***The classic GCN AGG$(\cdot)$ is consistently used, and layers formed by these three principles can be further stacked***. Other AGG$(\cdot)$ can be applied, but as long as they can achieve GCN, the introduced advan-

Table 2: Three IGNN variants with GCN AGG$(\cdot)$.

| | SN - $\mathbf{h}_v^{(k)}$ | IN - $\mathbf{m}_v^{(k)}$ | NR |
|---|---|---|---|
| **r-IGNN** | **No SN**. Coupled or shared $\mathbf{W}^{(k)}$. | $\sum \sigma(\widehat{\mathbf{A}}_{v,u}^k \mathbf{h}_u^{(k-1)})$ | $\mathbf{h}_v^{(k)} = \sigma(\mathbf{m}_v^{(k)}\mathbf{W}^{(k)}) + \mathbf{h}_v^{(k-1)}$ |
| **a-IGNN** | | | $\mathbf{h}_v^{(k)} = \alpha_v^{(k)}\mathbf{m}_v^{(k)} + (1-\alpha_v^{(k)})\mathbf{h}_v^{(k-1)}$ |
| **c-IGNN** | $\mathbf{x}_v \mathbf{W}^{(k)}$ | $\sum \sigma(\widehat{\mathbf{A}}_{v,u}^k \mathbf{h}_u^{(k)})$ | $\mathbf{h}_v = \sigma\left((\|_{i=0}^k \sigma(\mathbf{m}_v^{(i)}))\mathbf{W}\right)$ |

tages of IGNN always hold. Table 9 and 10 illustrates how existing works falls into IGNN variants.

**Residual r-IGNN** variants leverage the residual connection [45] as: $\mathbf{h}_v^{(k)} = \sigma(\mathbf{m}_v^{(k)}\mathbf{W}^{(k)}) + \mathbf{h}_v^{(k-1)}$, whose matrix format is $\mathbf{H}^{(k)} = \sigma(\widehat{\mathbf{A}}\mathbf{H}^{(k-1)}\mathbf{W}^{(k)}) + \mathbf{H}^{(k-1)}$. It is easy to observe that the expansion of $\mathbf{H}^{(k)}$ covers all $\widehat{\mathbf{A}}^i, 0 < i < k$ (see Appendix A.2), which is an inceptive variant with IN &NR designs. Besides, some methods [46, 16] adopt an initial residual connection, constructing connections to the initial representation $\mathbf{H}^{(0)}$ (see Appendix D.2). Luo et al. [11] *empirically* demonstrated that this variant equipped with dropout and batch normalization establishes a strong baseline, but the theoretical

rationale remains unclear. Our work extends this understanding by explaining its effectiveness under varying homophily through the lens of the smoothness-generalization dilemma. We first prove its adaptive smoothness capability in Theorem 5.1 and further expose its inherent generalization limitations via quantitative analysis in Section 5.2.3, thereby elucidating the necessity of dropout and batch normalization, which can improve generalization and prevent feature collapse [47].

**Attentive a-IGNN** variants leverage the attention mechanism to realize *node-wise personalized neighborhood relationship learning*, defined as: $\mathbf{h}_v^{(k)} = \alpha_v^{(k)}\mathbf{m}_v^{(k)} + (1 - \alpha_v^{(k)})\mathbf{h}_v^{(k-1)}$, where $\alpha_v^{(k)} = g^{(k)}(\mathbf{m}_v^{(k)}, \mathbf{h}_v^{(k-1)})$, and $g^{(k)}(\cdot)$ is the mechanism function. Several methods, such as DAGNN [48], GPRGNN [33], ACMGCN [24], and OrderedGNN [22], employ different attention mechanisms yet unintentionally share the same IN &NR design.

**Concatenative c-IGNN** variants concatenate multi-neighborhoods with a learnable transformation: $\mathbf{h}_v = \sigma\left((\|_{i=0}^{k}\sigma(\mathbf{m}_v^{(i)}))\mathbf{W}\right)$, where $\|$ means concatenation. A c-IGNN with GCN AGG($\cdot$) is $\mathbf{H}_{IG,k} = \sigma((\|_{i=0}^{k}\sigma(\widehat{\mathbf{A}}^i\mathbf{X}\mathbf{W}^{(i)}))\mathbf{W})$, $\mathbf{W}^{(i)} \in \mathbb{R}^{D \times F}$, and $\mathbf{W} \in \mathbb{R}^{kF \times F'}$. Although simple, its power is strong, as it can achieve various relationships, such as *general layer-wise neighborhood mixing*, *personalized* and *generalized PageRank* as in Proposition 5.2. Notably, when SN is incorporated in c-IGNN, the **REL**($\cdot$) becomes optional, as the SN and NR transformations can be merged.

## 5.2 Theoretical and Empirical Analysis of Dilemma Alleviation

### 5.2.1 IN &NR: Adaptive Smoothness Capabilities

**Theorem 5.1.** *Inceptive neighborhood relationship learning (IN &NR) can approximate arbitrary graph filters for adaptive smoothness capabilities extending beyond simple low- or high-pass ones, expressing the $K$ order polynimial graph filter ($\sum_{i=0}^{K}\theta_i\widehat{\mathbf{L}}^i$) with arbitrary coefficients $\theta_i$, including c-IGNN (SN, IN and NR), as well as r-IGNN and a-IGNN (IN &NR).*

**Proposition 5.2.** *IGNN-s can achieve (1) SIGN, (2) APPNP with personalized PageRank, (3) MixHop with general layerwise neighborhood mixing, and (4) GPRGNN with generalized PageRank.*

**Remark**. Wu et al. [49] found that the vanilla GCN essentially simulates a K-order polynomial filter [50] with *predetermined coefficients*, limited to a low-pass filter. However, many works has highlighted the significance of high-frequency signals for heterophily [24, 51]. *The inceptive neighborhood relationship learning module (IN +NR) benefits IGNN with the expressive power beyond simple low-pass or high-pass filters* as in Theorem 5.1, achieving the $K$-order polynomial graph filter with *arbitrary coefficients,* which has been proven able to approximate any graph filter [52]. Consequently, many existing methods are just simplified cases of IGNN as in Proposition 5.2.

### 5.2.2 SN: Improved Hop-wise and Overall Generalization

**Theorem 5.3.** *Let the representation of c-IGNN incorporating the SN principle be denoted as $\mathbf{H}_{IG,k} = \sigma((\|_{i=0}^{k}\sigma(\widehat{\mathbf{A}}^i\mathbf{X}\mathbf{W}^{(i)}))\mathbf{W})$, and the Lipschitz constant of it be denoted as $\hat{L}_{IG}$. Given $d_{\mathcal{M}}(\mathbf{X}) = \mathcal{D}$ and $\mathbf{W} = \begin{bmatrix}\mathbf{W}_0 \\ \cdots \\ \mathbf{W}_k\end{bmatrix}$, then the distance from $\mathbf{H}_{IG,k}$ to $\mathcal{M}$ satisfies:*

$$d_{\mathcal{M}}(\mathbf{H}_{IG,k}) \leq \left\|\sum_{i=0}^{k}\lambda^i\mathbf{W}^{(i)}\mathbf{W}_i\right\|_2 \mathcal{D}, \tag{4}$$

*where $\lambda < 1$ is the second largest eigenvalue of $\widehat{\mathbf{A}}$, and $\hat{L}_{IG} = \|\sum_{i=0}^{k}\mathbf{W}^{(i)}\mathbf{W}_i\|_2$.*

**Remark**. Theorem 5.3 demonstrates the mitigation of the dilemma from two perspectives. *From the **local perspective**, each $i$-th hop has a distinct Lipschitz constant with isolated transformations ($\mathbf{W}^{(i)}\mathbf{W}_i$), allowing for a separate handle of its own generalization expectations.* High-order homophilic neighborhoods with extremely small $\lambda^i$ demand large Lipschitz constants to mitigate massive information loss from oversmoothing, while low-order or heterophilic ones can enjoy small Lipschitz constants to guarantee good generalization. *From the **global perspective**, the entire network's Lipschitz constant is effectively shrunk from cascade multiplication to summation, avoiding the extreme decline in overall generalization ability*. The overall Lipschitz constant is a summation of individual multiplication of each layer transformation ($\hat{L}_{IG} = \|\sum_{i=0}^{k}\mathbf{W}^{(i)}\mathbf{W}_i\|_2$) in c-IGNN,

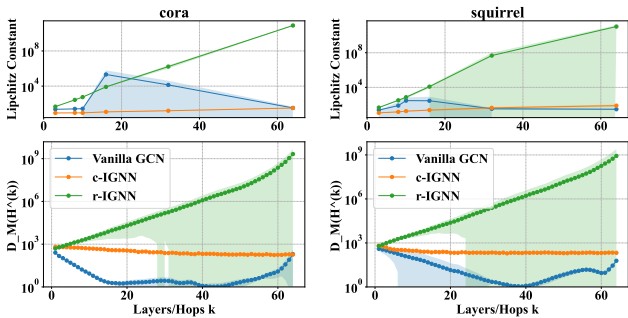

Figure 3: Quantitative Analysis on the Cora (Homophily) and Squirrel (Heterophily) Datasets.

whose increase in magnitude will be much smaller than that of cascade multiplication $\hat{L}_G = \|\prod_{i=0}^{k} \mathbf{W}^{(i)}\|_2$ in the traditional framework, which will grow exponentially as the layer increases since each high-order neighborhoods suffering from oversmoothing all demand large $\mathbf{W}^{(i)}$.

### 5.2.3 Quantitative Analysis on Smoothness-Generalization Delimma

We conducted a quantitative study of the dilemma using three GNNs on the Cora and Squiirel dataset: (1) vanilla GCN, (2) r-IGNN (**IN** and **NR**), and (3) c-IGNN (**IN**, **NR**, and **SN**). The trends of $d_{\mathcal{M}}(\mathbf{H}^{(k)})$ and Lipschitz constant $\hat{L}$, computed following Cong et al. [53], are presented in Figure 3.

**First**, as $k$ increases in vanilla GCN, $d_{\mathcal{M}}(\mathbf{H}^{(k)})$ initially decreases (indicating increased smoothness) before rising again due to strong supervision from the classifier. In contrast, $\hat{L}$ follows an inverse pattern. *This behavior aligns with the smoothness–generalization dilemma.* **Second**, while r-IGNN alleviates oversmoothing, as evidenced by the increased $d_{\mathcal{M}}(\mathbf{H}^{(k)})$, *it exhibits a steadily increasing $\hat{L}$, suggesting degraded generalization.* **Finally**, c-IGNN, which integrates all three principles, demonstrates stable and moderate trends in both $d_{\mathcal{M}}(\mathbf{H}^{(k)})$ and $\hat{L}$, indicating *its ability to preserve generalization while avoiding excessive smoothness.* See Appendix C for more details.

## 6 Experiments

Research questions are: **RQ1**: How does IGNN perform compared to SOTA methods? **RQ2**: What are the contributions of the three principles? **RQ3**: How is the dilemma resolved across various hops?

### 6.1 Datasets, Baselines and Settings

**Datasets**: Following recent works [54], we select 13 representative datasets of various sizes, excluding those too small or class-imbalanced [27]: *(i) Heterophily*: Roman-empire, BlogCatalog, Flickr, Actor, Squirrel-filtered, Chameleon-filtered, Amazon-ratings, Pokec; *(ii) Homophily*: PubMed, Photo, wikics, ogbn-arxiv, ogbn-products. The statistics are in Table 3 and 4.

**Baselines**: We selected 30 representative baselines, as shown in Table 11. These models are categorized into four types: graph-agnostic models, homophilic GNNs, heterophilic GNNs, and graph transformers. GNNs are further divided into Non-inceptive and Inceptive ones.

**Settings**: We randomly construct 10 splits with proportions of 48%/32%/20% for training/valida-tion/testing, which is guided by our theoretical emphasis on generalization. Prior work [40] has shown that different splitting strategies can lead to substantial variations in structural distributions, thereby influencing generalization behavior. To mitigate this, we adopt a unified split scheme [19, 22], reducing variance across datasets that may arise from the heterogeneous splitting policies used in earlier studies. For the large-size datasets (ogbn-arxiv, Pokec, and ogbn-products), we use the public splits. The network is optimized using the Adam [55], with hyperparameter settings provided in Appendix E.2. Our code with best hyperparamter settings and search scripts are available at https://github.com/galogm/IGNN. Additional results and code for *public splits* are also provided in the repository. We report the mean and standard deviation of classification accuracy across splits, *with complexity, paramter count and runtime analysis and comparison* documented in Appendix B.

Table 3: Overall Performance of Node Classification. The best results are in **bold**, and the second-best results are underlined. A.R is the average of all ranks across datasets. OOM means out of memory.

| | | Dataset | Actor | Blog | Flickr | Roman-E | Squirrel-f | Chame-f | Amazon-R | Pubmed | Photo | Wikics | |
|---|---|---|---|---|---|---|---|---|---|---|---|---|---|
| | | $h_e$ | 0.2163 | 0.4011 | 0.2386 | 0.0469 | 0.2072 | 0.2361 | 0.3804 | 0.8024 | 0.8272 | 0.6543 | |
| | | #Nodes | 7,600 | 5,196 | 7,575 | 22,662 | 2,223 | 890 | 24,492 | 19,717 | 7,650 | 11,701 | A.R. |
| | | #Edges | 33,544 | 171,743 | 239,738 | 32,927 | 46,998 | 8,854 | 93,050 | 44,338 | 238,162 | 431,206 | |
| | | #Feats | 931 | 8,189 | 12,047 | 300 | 2,089 | 2,325 | 500 | 500 | 745 | 300 | |
| | | MLP | 34.69±0.71 | 93.08±0.63 | 89.41±0.73 | 62.12±1.79 | 34.00±2.44 | 35.00±3.29 | 42.25±0.73 | 87.68±0.51 | 86.73±2.20 | 73.51±1.18 | 29.5 |
| Homophilic | Non. | SGC | 29.46±0.96 | 72.85±1.15 | 59.02±1.48 | 42.90±0.50 | 39.75±1.85 | 42.42±3.28 | 41.32±0.80 | 87.14±0.57 | 92.38±0.49 | 77.63±0.88 | 27.2 |
| | | GCN | 30.82±1.41 | 77.28±1.43 | 69.06±1.70 | 36.23±0.57 | 37.06±1.42 | 41.46±3.42 | 44.96±0.40 | 87.70±0.58 | 94.88±2.08 | 78.59±1.07 | 26.7 |
| | | GAT | 30.94±0.95 | 85.36±1.37 | 57.87±2.22 | 62.31±0.93 | 34.22±1.41 | 40.69±3.20 | 47.41±0.80 | 87.64±0.54 | 94.72±0.52 | 76.92±0.81 | 28.0 |
| | | GraphSAGE | 34.52±0.64 | 95.73±0.53 | 91.74±0.58 | 66.39±2.16 | 34.83±2.24 | 41.24±1.65 | 46.71±2.83 | 88.71±0.65 | 94.52±1.27 | 80.85±1.00 | 23.2 |
| | Incep. | APPNP | 35.09±0.79 | 96.13±0.58 | 91.21±0.52 | 71.76±0.34 | 34.18±1.68 | 41.12±3.25 | 47.72±0.54 | 87.97±0.62 | 95.05±0.43 | 83.04±0.94 | 21.9 |
| | | JKNet-GCN | 30.49±1.71 | 84.25±0.71 | 71.72±1.47 | 69.61±0.42 | 40.11±2.54 | 43.31±3.12 | 48.15±0.93 | 87.41±0.38 | 94.39±0.40 | 83.80±0.65 | 23.0 |
| | | IncepGCN | 35.69±0.75 | 96.67±0.48 | 90.42±0.71 | 80.97±0.49 | 38.27±1.36 | 43.31±2.18 | 52.72±0.80 | 89.32±0.47 | 95.66±0.40 | 85.22±0.48 | 12.0 |
| | | SIGN | 36.76±1.00 | 96.06±0.68 | 91.81±0.58 | 81.56±0.57 | 42.13±1.99 | 44.66±3.46 | 52.47±0.95 | 90.29±0.50 | 95.53±0.43 | 85.59±0.79 | 7.7 |
| | | MixHop | 36.82±0.98 | 96.05±0.48 | 89.78±0.63 | 79.39±0.40 | 41.35±1.04 | 44.61±3.16 | 47.91±0.53 | 89.40±0.37 | 94.91±0.45 | 83.15±0.96 | 15.8 |
| | | FAGCN | 35.98±1.34 | 96.67±0.35 | 92.74±0.79 | 75.65±1.01 | 40.83±3.08 | 42.70±3.33 | 50.14±0.76 | 90.24±0.51 | 95.31±0.45 | 85.02±0.51 | 10.6 |
| | | ωGAT | 34.66±0.97 | 94.95±0.61 | 90.20±1.13 | 80.98±1.00 | 34.07±2.16 | 41.07±4.23 | 48.81±0.92 | 89.58±0.50 | 95.19±0.47 | 85.17±0.83 | 19.1 |
| | | DAGNN | 35.04±1.03 | 96.73±0.61 | 92.18±0.73 | 73.94±0.45 | 35.62±1.48 | 40.96±2.91 | 50.44±0.52 | 89.76±0.55 | 95.70±0.40 | 85.07±0.73 | 14.2 |
| | | GCNII | 35.69±1.08 | 96.25±0.61 | 91.36±0.68 | 80.55±0.82 | 38.43±2.10 | 42.13±2.04 | 47.65±0.48 | 90.00±0.46 | 95.54±0.34 | 85.15±0.56 | 13.7 |
| Heterophilic | Non. | H2GCN | 32.74±1.23 | 96.32±0.62 | 91.33±0.59 | 68.70±1.66 | 33.89±1.01 | 38.09±2.63 | 36.65±0.73 | 89.50±0.43 | 91.56±1.49 | 74.76±3.39 | 25.5 |
| | | GBKGNN | 35.74±4.46 | OOM | OOM | 66.10±4.61 | 34.58±1.63 | 41.52±2.36 | 41.00±1.62 | 88.66±0.43 | 93.39±2.00 | 81.85±1.83 | 26.7 |
| | | GGCN | 35.72±1.48 | 96.09±0.55 | 90.17±0.76 | OOM | 36.04±2.61 | 38.54±3.99 | OOM | 89.19±0.43 | 95.32±0.27 | 83.67±0.75 | 23.1 |
| | | GloGNN | 35.82±1.27 | 92.53±0.80 | 88.18±0.85 | 70.87±0.89 | 35.39±1.70 | 40.28±2.91 | 49.01±0.74 | 88.14±0.25 | 92.15±0.33 | 84.20±0.55 | 23.6 |
| | | HOGGCN | 36.05±1.06 | 95.79±0.59 | 90.40±0.64 | OOM | 35.10±1.81 | 38.43±3.66 | OOM | OOM | 94.48±0.50 | 83.57±0.63 | 25.5 |
| | Incep. | GPRGNN | 35.79±1.04 | 96.26±0.62 | 91.52±0.56 | 72.36±0.38 | 38.00±1.58 | 41.63±2.86 | 46.07±0.78 | 89.45±0.61 | 95.51±0.39 | 83.16±1.23 | 17.6 |
| | | ACMGCN | 35.68±1.17 | 96.01±0.53 | 68.63±1.87 | 72.58±0.35 | 37.60±1.70 | 43.03±3.08 | 50.51±0.66 | 89.95±0.50 | 92.35±0.39 | 84.13±0.66 | 19.1 |
| | | OrderedGNN | 36.95±0.85 | 96.39±0.69 | 91.13±0.59 | 82.65±0.91 | 36.27±1.95 | 42.13±3.04 | 51.58±0.99 | 90.01±0.40 | 95.87±0.24 | 85.60±0.77 | 9.9 |
| | | N² | 37.41±0.60 | 94.72±0.57 | 91.08±0.79 | 75.32±0.41 | 39.35±2.39 | 38.60±1.12 | 48.08±0.76 | 89.16±0.24 | 95.92±0.27 | 84.07±0.39 | 16.4 |
| | | CoGNN | 37.52±1.66 | 96.41±0.56 | 89.91±0.93 | 87.57±0.46 | 37.89±2.23 | 40.45±2.48 | 52.89±0.81 | 89.49±0.53 | 95.15±0.55 | 85.70±0.71 | 12.6 |
| | | UniFilter | 36.11±1.04 | 96.53±0.47 | 91.89±0.75 | 74.90±0.91 | 42.40±2.58 | 46.07±4.74 | 49.36±0.98 | 90.15±0.39 | 94.91±0.62 | 85.43±0.67 | 9.8 |
| | | NodeFormer | 36.10±1.09 | 94.28±0.67 | 89.05±0.99 | 70.24±1.58 | 38.38±1.81 | 38.93±3.68 | 42.67±0.77 | 88.36±0.43 | 93.81±0.75 | 80.98±0.84 | 23.7 |
| | | DIFFormer | 36.13±1.19 | 96.50±0.71 | 90.86±0.58 | 79.36±0.54 | 41.12±1.09 | 41.69±2.96 | 49.33±0.97 | 88.90±0.47 | 95.67±0.29 | 84.27±0.75 | 13.6 |
| | | SGFormer | 37.36±1.11 | 96.98±0.59 | 91.62±0.55 | 75.71±0.44 | 42.22±2.45 | 44.44±3.01 | 51.60±0.62 | 89.75±0.44 | 95.84±0.41 | 84.72±0.72 | 8.4 |
| | | GOAT | 35.90±1.31 | 95.20±0.54 | 89.43±1.28 | 79.41±0.81 | 36.27±2.13 | 44.10±4.06 | 51.47±0.96 | 89.85±0.57 | 95.48±0.33 | 85.56±0.72 | 14.3 |
| | | Polynormer | 37.27±1.52 | 96.73±0.45 | 91.98±0.74 | **92.46±0.43** | 40.13±2.28 | 43.60±3.29 | 53.35±1.06 | 89.98±0.44 | 95.75±0.22 | 84.76±0.82 | 6.5 |
| Ours | Incep. | r-IGNN | 37.58±1.39 | 96.49±0.39 | 92.32±0.66 | 90.36±0.43 | 44.67±2.08 | 46.63±3.30 | 52.10±1.02 | 89.76±0.49 | 95.53±0.42 | 85.20±0.61 | 6.5 |
| | | a-IGNN | 38.04±1.00 | 96.77±0.42 | 93.24±0.73 | 90.96±0.53 | 45.01±2.65 | 47.53±3.09 | 52.22±0.66 | 90.22±0.52 | 95.73±0.38 | 85.75±0.59 | 3.2 |
| | | c-IGNN | **38.51±0.94** | **97.24±0.34** | **93.27±0.40** | 90.97±0.36 | **45.71±2.13** | **50.79±4.92** | 53.03±0.61 | **90.41±0.59** | 95.91±0.29 | **86.37±0.44** | **1.3** |

## 6.2 Performance Analysis (RQ1)

From Table 3 and 4, it is evident that IGNN incorporating all three principles consistently outperforms baselines.

*A subset of homoGNNs, which happen to be inceptive variants, outperform many recent heteroGNNs, highlighting the strength of inceptive architectures in addressing the dilemma hindering universality.* Specifically, the average ranks of inceptive homoGNNs exceed those of all non-inceptive heteroGNNs, and in many cases, surpass those of inceptive heteroGNNs. These homoGNNs have been largely overlooked previously, as their designs are not tailored for heterophily. Only DAGNN and GCNII have specific features to mitigate oversmoothing. Surprisingly, the mere incorporation of inceptive designs is sufficient to achieve superior performance. This strongly suggests that the key factor limiting universality is the dilemma.

Table 4: Performance on Large Datasets.

| Dataset | ogbn-arxiv | pokec | ogbn-products |
|---|---|---|---|
| $h_e$ | 0.66 | 0.44 | 0.81 |
| #Nodes | 169,343 | 1,632,803 | 2,440,029 |
| #Edges | 1,166,243 | 30,622,564 | 123,718,280 |
| #Feats | 128 | 65 | 100 |
| MLP | 55.50±0.23 | 63.27±0.12 | 61.06±0.12 |
| GCN | 71.74±0.29 | 74.45±0.27 | 75.45±0.16 |
| GAT | 71.74±0.29 | 72.77±3.18 | 79.45±0.28 |
| SGC | 70.74±0.29 | 73.77±3.18 | 74.78±0.17 |
| SIGN | 70.28±0.25 | 77.98±0.14 | 77.60±0.13 |
| GPRGNN | 71.40±0.32 | 78.62±0.15 | 78.23±0.25 |
| NodeFormer | 67.72±0.52 | 70.12±0.42 | 71.23±1.40 |
| DIFFormer | 69.85±0.34 | 72.89±0.56 | 74.16±0.32 |
| SGFormer | 72.62±0.18 | 73.24±0.54 | 76.24±0.45 |
| r-IGNN | 72.63±0.23 | **82.74±0.41** | 80.92±0.19 |
| a-IGNN | 72.60±0.31 | 82.09±0.25 | 78.89±0.47 |
| c-IGNN | **73.26±0.10** | 82.09±0.11 | **82.04±0.45** |

*Inceptive heteroGNNs demonstrate better performance compared to non-inceptive heteroGNNs, while graph transformers also show relatively strong performance.* First, inceptive heteroGNNs are mostly attentive variants employing different attention mechanisms. Interestingly, these models exhibit significant differences in performance, indicating that the design of the attention mechanism plays a critical role. Second, graph transformers excel likely because they move beyond the traditional message passing process, which utilizes the global attention mechanisms. Notably, Polynormer shows a great advantage on roman-empire which is not observed in other datasets. Upon examination, we found it was a long-chain graph derived from words, aligning with the inherent strengths of transformers in natural language processing. Nevertheless, we observe an interesting ***insight for language graphs***: for the same receptive field size $k$, they achieve better performance when stacking $k$ IGNN layers than when using a single IGNN layer with RN across $k$ hops. As we focus on general graphs and the A.R. of IGNN-s show consistent advantages, we leave such graphs to future studies.

*IGNN outperforms all baselines with or without inceptive architectures, while inceptive GNNs also vary in performance, suggesting that the effectiveness is significantly influenced by whether all principles are integrated and how they are implemented.* In particular, concatenative variants (e.g., c-IGNN, SIGN, and IncepGCN) generally outperform residual and attentive ones, with the ordered

Table 5: Ablation of Three Principles. *A.R.* denotes the average of all ranks across datasets.

| | GCN AGG(·)+ | | | Equivalent Variant | Actor | Blog | Flickr | Roman-E | Squirrel-f | Chame-f | Amazon-R | Pubmed | Photo | Wikics | A.R. |
|---|---|---|---|---|---|---|---|---|---|---|---|---|---|---|---|
| | SN | IN | NR | | | | | | | | | | | | |
| 1 | | | | GCN | 30.82±1.41 | 77.28±1.43 | 69.06±1.70 | 36.23±0.57 | 37.06±1.42 | 41.46±3.42 | 44.96±0.40 | 87.70±0.58 | 94.88±2.08 | 78.59±1.07 | 5.7 |
| 2 | | ✓ | | SIGN w/o SN | 36.32±1.03 | 96.89±0.29 | 91.81±0.76 | 79.77±0.95 | 42.52±2.52 | 44.10±4.24 | 51.72±0.69 | 89.63±0.54 | 95.74±0.41 | 85.67±0.70 | 3.2 |
| 3 | | | ✓ | JKNet-GCN | 30.49±1.71 | 84.25±0.71 | 71.72±1.47 | 69.61±0.42 | 40.11±2.54 | 43.31±3.12 | 48.15±0.93 | 87.41±0.38 | 94.39±0.40 | 83.80±0.65 | 5.3 |
| 4 | ✓ | ✓ | | SIGN | 36.76±1.00 | 96.06±0.68 | 91.81±0.58 | 81.56±0.57 | 42.13±1.99 | 44.66±3.46 | 52.47±0.95 | 90.29±0.50 | 95.53±0.43 | 85.59±0.79 | 3.0 |
| 5 | | ✓ | ✓ | r-IGNN | 37.58±1.39 | 96.49±0.39 | 92.32±0.66 | 90.36±0.43 | 44.67±2.08 | 46.63±3.80 | 52.10±1.02 | 89.76±0.49 | 95.53±0.42 | 85.20±0.61 | 2.6 |
| 6 | ✓ | ✓ | ✓ | c-IGNN | **38.51±0.94** | **97.24±0.34** | **93.27±0.40** | **90.97±0.36** | **45.71±2.13** | **50.79±4.92** | **53.03±0.61** | **90.41±0.59** | **95.91±0.29** | **86.37±0.44** | **1.0** |

gating mechanism of OrderedGNN standing out as evidence that order information is crucial for capturing neighborhood relationships. However, two concatenative variants show low performance due to unique designs: original JKNet does not include ego features without propagation, and MixHop requires stacking layers, reintroducing transforamtion decoupling. Furthermore, most inceptive GNNs fail to incorporate all three principles, thereby not fully resolving the dilemma and degrading their performance on universality. See a detailed comparison of inceptive GNNs in Appendix D.1

### 6.3 Ablation Studies of SN, IN and NR (RQ2)

Table 5 presents the ablation of the three principles. It is important to note that SN cannot be applied without IN, so the ablations do not include any combinations of SN without IN. Several key conclusions can be drawn: *First*, the best performance is achieved when all principles are applied, as c-IGNN obtains the highest average rank (Rank 1) (line 6 vs. others). *Second*, JKNet-GCN shows a significant performance gap depending on IN (line 3 vs. line 5), where the difference lies in whether each hop is aggregated independently with the ego feature transformation included. This indicates that incorporating IN and the ego representation into the final representation enhances generalization. *Third*, SN and NR demonstrate excellent synergy, yielding significantly improved results when used together. Although IN is incorpo-

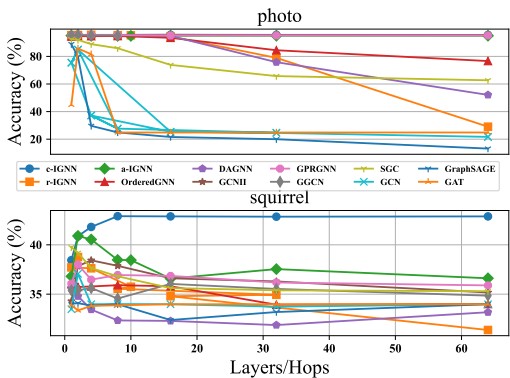

Figure 4: Performance of Different Hops

rated in lines 4–6, adding either SN or NR alone (lines 4, 5) does not lead to the best improvement compared to incorporating both, as seen in c-IGNN (line 6).

### 6.4 Performance of Different Neighborhood Hops (RQ3)

Figure 4 illustrates various method performance across different hops. In the homophilic context (photo), many inceptive methods effectively mitigating the oversmoothing issue, such as GCNII, GPRGNN, IGNN and OrderedGNN. Conversely, in the heterophilic scenario (squirrel), most of them consistently struggle with high-order neighborhoods, as evidenced by a trend of initial improvement followed by a decline in performance. In contrast, c-IGNN exhibits a notable increase in performance that stabilizes thereafter, highlighting the effectiveness of incorporating all three principles in improving hop-wise and overall generalization as well as alliviating the dilemma.

## 7 Conclusion

This paper advances GNN universality across varying homophily by identifying the smoothness-generalization dilemma, which impairs learning in high-order homophilic neighborhoods and all heterophilic ones. We propose the Inceptive Graph Neural Network (IGNN), a unified message-passing framework built on three key design principles: separative neighborhood transformation, inceptive neighborhood aggregation, and neighborhood relationship learning. These principles alleviate the dilemma by enabling distinct hop-wise generalization, improving overall generalization, and approximating arbitrary graph filters for adaptive smoothness. Extensive benchmarking against 30 baselines demonstrates IGNN 's superiority and reveals notable universality in certain homophilic GNN variants. For limitation discussion, please refer to Appendix F.

## Acknowledgments and Disclosure of Funding

This work is supported by the National Natural Science Foundation of China (Grant No. 62476245), Zhejiang Provincial Natural Science Foundation of China (Grant No. LTGG23F030005).

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

# A  Proofs of Theoretical Results

## A.1  Proofs of Theorems 4.1 and Corollary 4.2

**Restatement of Theorem 4.1.** *Given a graph $\mathcal{G}(\mathbf{X}, \mathbf{A})$, let the representation obtained via $k$ rounds of GCN message passing on symmetrically normalized $\widehat{\mathbf{A}}$ be denoted as $\mathbf{H}_G^{(k)} = \sigma(\widehat{\mathbf{A}}\mathbf{H}^{(k-1)}\mathbf{W}^{(k)})$, and the Lipschitz constant of this $k$-layer graph neural network be denoted as $\hat{L}_G$. Given the distance from $\mathbf{X}$ to the subspace $\mathcal{M}$ as $d_{\mathcal{M}}(\mathbf{X}) = \mathcal{D}$, then the distance from $\mathbf{H}_G^{(k)}$ to $\mathcal{M}$ satisfies:*

$$d_{\mathcal{M}}(\mathbf{H}_G^{(k)}) \leq \hat{L}_G \lambda^k \mathcal{D}, \tag{5}$$

*where $\hat{L}_G = \| \prod_{i=0}^{k} \mathbf{W}^{(i)} \|_2$, and $\lambda < 1$ is the second largest eigenvalue of $\widehat{\mathbf{A}}$.*

*proof of Theorem 4.1.* To prove Theorem 4.1, we need to borrow the following notations and Lemmas from Oono and Suzuki [29]. For $N, D, F \in \mathbb{N}_+$, $\widehat{\mathbf{A}} \in \mathbb{R}^{N \times N}$ is a symmetric matrix and $\mathbf{W}^{(k)} \in \mathbb{R}^{D \times F}$ for $k \in \mathbb{N}_+$. For $M \leq N$, let $\mathbf{U}$ be a $M$-dimensional subspace of $\mathbb{R}^N$. We assume $\mathbf{U}$ and $\widehat{\mathbf{A}}$ satisfy the following properties that generalize the situation where $\mathbf{U}$ is the eigenspace associated with the smallest eigenvalue of the graph Laplacian $\widehat{\mathbf{L}} = \mathbf{I}_N - \widehat{\mathbf{A}}$ (that is, zero). We endow $\mathbb{R}^N$ with the ordinal inner product and denote the orthogonal complement of $\mathbf{U}$ by $\mathbf{U}^{\perp} := \{ \mathbf{u} \in \mathbb{R}^N \mid \langle \mathbf{u}, \mathbf{v} \rangle = 0, \forall \mathbf{v} \in \mathbf{U} \}$. We can regard $\widehat{\mathbf{A}}$ as a linear mapping $\widehat{\mathbf{A}}\big|_{\mathbf{U}^{\perp}} : \mathbf{U}^{\perp} \to \mathbf{U}^{\perp}$. Choose the orthonormal basis $(e_m)_{m=M+1,\dots,N}$ of $U^{\perp}$ consisting of the eigenvalue of $\widehat{\mathbf{A}}\big|_{U^{\perp}}$. Let $\lambda_m$ be the eigenvalue of $\widehat{\mathbf{A}}$ to which $e_m$ is associated ($m = M + 1, \dots, N$). Note that since the operator norm of $\widehat{\mathbf{A}}\big|_{\mathbf{U}^{\perp}}$ is $\lambda$, we have $|\lambda_m| \leq \lambda$ for all $m = M + 1, \dots, N$. Since $(e_m)_{m \in [N]}$ forms the orthonormal basis of $\mathbb{R}^N$, we can uniquely write $\mathbf{X} \in \mathbb{R}^{N \times D}$ as $\mathbf{X} = \sum_{m=1}^{N} e_m \otimes \boldsymbol{\omega}_m$ for some $\boldsymbol{\omega}_m \in \mathbb{R}^D$ with $\otimes$ denoting the Kronecker product. Then, we have

$$d_{\mathcal{M}}^2(\mathbf{X}) = \sum_{m=M+1}^{N} \|\boldsymbol{\omega}_m\|_2^2, \tag{6}$$

where $\|\cdot\|_2$ is the 2-norm. On the other hand, we have

$$
\begin{aligned}
\widehat{\mathbf{A}}\mathbf{X}\mathbf{W}^{(k)} &= \sum_{m=1}^{N} (\widehat{\mathbf{A}}e_m) \otimes (\mathbf{W}^{(k)^\top}\boldsymbol{\omega}_m) \\
&= \sum_{m=1}^{M} (\widehat{\mathbf{A}}e_m) \otimes (\mathbf{W}^{(k)^\top}\boldsymbol{\omega}_m) + \sum_{m=M+1}^{N} (\widehat{\mathbf{A}}e_m) \otimes (\mathbf{W}^{(k)^\top}\boldsymbol{\omega}_m) \qquad (7) \\
&= \sum_{m=1}^{M} (\widehat{\mathbf{A}}e_m) \otimes (\mathbf{W}^{(k)^\top}\boldsymbol{\omega}_m) + \sum_{m=M+1}^{N} e_m \otimes (\lambda_m \mathbf{W}^{(k)^\top}\boldsymbol{\omega}_m).
\end{aligned}
$$

Since $\mathbf{U}$ is invariant under $\widehat{\mathbf{A}}$ [29], for any $m \in [M]$, we can write $\widehat{\mathbf{A}}e_m$ as a linear combination of $e_n (n \in [M])$. Therefore, we have

$$
d_{\mathcal{M}}^2\left(\widehat{\mathbf{A}}\mathbf{X}\mathbf{W}^{(k)}\right) = \sum_{m=M+1}^{N} \left\|\lambda_m \mathbf{W}^{(k)^\top}\boldsymbol{\omega}_m\right\|_2^2. \qquad (8)
$$

**Lemma A.1** (Oono and Suzuki [29]). *For any $\mathbf{X} \in \mathbb{R}^{N \times D}$, we have $d_{\mathcal{M}}(\sigma(\mathbf{X})) \leq d_{\mathcal{M}}(\mathbf{X})$.*

Based on Lemma A.1, by simplifying the GCNs by removing the nonlinear activation functions in the intermediate layers [56, 49, 57] and retaining only the final activation function, we have

$$
\begin{aligned}
d_{\mathcal{M}}^2\left(\mathbf{H}_{\widehat{\mathbf{A}}}^{(k)}\right) &= d_{\mathcal{M}}^2\left(\sigma(\widehat{\mathbf{A}}\mathbf{H}_{\widehat{\mathbf{A}}}^{(k-1)}\mathbf{W}^{(k)})\right) \\
&\leqslant d_{\mathcal{M}}^2\left(\widehat{\mathbf{A}}\mathbf{H}_{\widehat{\mathbf{A}}}^{(k-1)}\mathbf{W}^{(k)}\right) \\
&= d_{\mathcal{M}}^2\left(\widehat{\mathbf{A}}^2 \mathbf{H}_{\widehat{\mathbf{A}}}^{(k-2)}\mathbf{W}^{(k-1)}\mathbf{W}^{(k)}\right) \qquad (9) \\
&= d_{\mathcal{M}}^2\left(\widehat{\mathbf{A}}^k \mathbf{X}\mathbf{W}^{(1)}\mathbf{W}^{(2)}\dots\mathbf{W}^{(k)}\right) \\
&= \sum_{m=M+1}^{N} \left\|\lambda_m^k\left(\mathbf{W}^{(1)}\dots\mathbf{W}^{(k)}\right)^\top \boldsymbol{\omega}_m\right\|_2^2.
\end{aligned}
$$

**Lemma A.2** (Juvina et al. [58]). *For any $k$-layer GCN with 1-Lipschitz activation functions (e.g. ReLU, Leaky ReLU, SoftPlus, Tanh or Sigmoid), defined as $\mathbf{H}^{(k)} = \sigma(\widehat{\mathbf{A}}\mathbf{H}^{(k-1)}\mathbf{W}^{(k)})$, the Lipschitz constant becomes*

$$
\hat{L}_G = \left\|\prod_{i=1}^{k}\mathbf{W}^{(i)}\right\|_2. \qquad (10)
$$

We recall the the Lipschitz constant $\hat{L}_G$ of GCN [58] as in Lemma A.2, and substitute Equation (10) into Equation (9), we have:

$$
\begin{aligned}
d_{\mathcal{M}}^2\left(\mathbf{H}_{\widehat{\mathbf{A}}}^{(k)}\right) &\leqslant \sum_{m=M+1}^{N} \left\|\lambda_m^k\left(\mathbf{W}^{(1)}\dots\mathbf{W}^{(k)}\right)^\top \boldsymbol{\omega}_m\right\|_2^2 \\
&\leqslant \sum_{m=M+1}^{N} \lambda_m^{2k}\|\boldsymbol{\omega}_m\|_2^2 \left\|\prod_{i=1}^{k}\mathbf{W}^{(i)}\right\|_2^2 \qquad (11) \\
&= \hat{L}_G^2 \sum_{m=M+1}^{N} \lambda_m^{2k}\|\boldsymbol{\omega}_m\|_2^2 \\
&\leqslant \hat{L}_G^2 \lambda^{2k} \sum_{m=M+1}^{N} \|\boldsymbol{\omega}_m\|_2^2 \qquad = \hat{L}_G^2 \lambda^{2k} d_{\mathcal{M}}^2(\mathbf{X}).
\end{aligned}
$$

$\square$

**Restatement of Corollary 4.2.** $\forall \hat{L}_G, \epsilon > 0, \exists k^* = \lceil (\log \frac{\epsilon}{\hat{L}_G \mathcal{D}})/\log \lambda \rceil$, such that $d_{\mathcal{M}}(\mathbf{H}_G^{(k^*)}) < \epsilon$, where $\lceil \cdot \rceil$ is the ceil of the input.

*proof of Corollary 4.2.* In order to have $d_{\mathcal{M}}(\mathbf{H}_{\widehat{\mathbf{A}}}^{(k)}) \leq \hat{L}_G \lambda^k \mathcal{D} < \epsilon$, since $\hat{L}_G >= 0$, $\mathcal{D} >= 0$ and $\lambda < 1$, we have

$$d_{\mathcal{M}}(\mathbf{H}_{\widehat{\mathbf{A}}}^{(k)}) \leq \hat{L}_G \lambda^k \mathcal{D} < \epsilon \Rightarrow \lambda^k < \frac{\epsilon}{\hat{L}_G \mathcal{D}},$$
$$\Rightarrow k \log \lambda < \log \frac{\epsilon}{\hat{L}_G \mathcal{D}}, \tag{12}$$
$$\Rightarrow k > \frac{\log \frac{\epsilon}{\hat{L}_G \mathcal{D}}}{\log \lambda}.$$

Therefore, there exists $k^* = \lceil \frac{\log \frac{\epsilon}{\hat{L}_G \mathcal{D}}}{\log \lambda} \rceil$, such that $d_{\mathcal{M}}(\mathbf{H}_{\widehat{\mathbf{A}}}^{(k^*)}) \leq \hat{L}_G \lambda^{k^*} \mathcal{D} < \epsilon$, where $\lceil \cdot \rceil$ is the ceil of the input. $\square$

## A.2 Proof of Theorem 5.1

In this subsection, we present the proofs for the concatenative (c-IGNN), residual (r-IGNN), and attentive (a-IGNN) variants, demonstrating their expression capability of the K-order polynomial graph filter with arbitrary coefficients.

**Restatement of Theorem 5.1.** *Inceptive neighborhood relationship learning (IN &NR) can approximate arbitrary graph filters for adaptive smoothness capabilities* extending beyond simple low- or high-pass ones, expressing the $K$ order polynimial graph filter ($\sum_{i=0}^{K} \theta_i \widehat{\mathbf{L}}^i$) with arbitrary coefficients $\theta_i$ , including **c-IGNN** (SN, IN and NR), as well as **r-IGNN** and **a-IGNN** (IN &NR).

*Proof of the Concatenative Variant **c-IGNN**.* A polynomial graph filter [50] defined on $\widehat{\mathbf{A}}$ is given by:

$$\mathbf{H}_p = \left( \sum_{k=0}^{K} \theta_k \widehat{\mathbf{L}}^k \right) \mathbf{X} = \left( \sum_{k=0}^{K} \theta_k (\mathbf{I}_N - \widehat{\mathbf{A}})^k \right) \mathbf{X}. \tag{13}$$

Expanding $(\mathbf{I}_N - \widehat{\mathbf{A}})^k$ using the binomial theorem and rearranging the summation order yields:

$$\mathbf{H}_p = \left( \sum_{k=0}^{K} \theta_k \left( \sum_{i=0}^{k} (-1)^i \binom{k}{i} \widehat{\mathbf{A}}^i \right) \right) \mathbf{X} = \left( \sum_{i=0}^{K} \left( \sum_{k=i}^{K} \theta_k (-1)^i \binom{k}{i} \widehat{\mathbf{A}}^i \right) \right) \mathbf{X}. \tag{14}$$

Meanwhile, the matrix formulation of c-IGNN can be expressed as:

$$\mathbf{H} = \sigma \left( \left( \overset{K}{\underset{k=0}{\big\|}} \sigma(\widehat{\mathbf{A}}^k \mathbf{X} \mathbf{W}^{(k)}) \right) \mathbf{W} \right) = \sigma \left( \sum_{k=0}^{K} \sigma(\widehat{\mathbf{A}}^k \mathbf{X} \mathbf{W}^{(k)}) \mathbf{W}_k \right), \tag{15}$$

where $\mathbf{W} = \begin{bmatrix} \mathbf{W}_0 \\ \cdots \\ \mathbf{W}_k \\ \cdots \\ \mathbf{W}_K \end{bmatrix}$. By simplifying the above expression, omitting the non-linear layers, and setting $\mathbf{W}^{(k)} = \mathbf{I}$, $\mathbf{W}_k = (\sum_{i=k}^{K} \theta_i (-1)^k \binom{i}{k})\mathbf{I}$, we obtain:

$$\mathbf{H} = \sum_{k=0}^{K} (\widehat{\mathbf{A}}^k \mathbf{X} \mathbf{I})(\sum_{i=k}^{K} \theta_i (-1)^k \binom{i}{k})\mathbf{I} = \sum_{k=0}^{K} \sum_{i=k}^{K} \theta_i (-1)^k \binom{i}{k} \widehat{\mathbf{A}}^k \mathbf{X}. \tag{16}$$

Swapping the notation of $i$ and $k$, we get $\mathbf{H} = \sum_{i=0}^{K} \sum_{k=i}^{K} \theta_k (-1)^i \binom{k}{i} \widehat{\mathbf{A}}^i \mathbf{X}$, which matches the polynomial graph filter form in Equation (13). Since coefficients $(\sum_{i=k}^{K} \theta_i (-1)^k \binom{i}{k})$ can be arbitrary to learn by each $\mathbf{W}_k$, the concatenative variant (c-IGNN) is capable of expressing the K-order polynomial graph filter with arbitrary coefficients. $\square$

*Proof of the Residual Variant **r-IGNN***. We begin by verifying, using mathematical induction, that the residual variant $\mathbf{H}^{(k)} = \widehat{\mathbf{A}}\mathbf{H}^{(k-1)}\mathbf{W}^{(k)} + \mathbf{H}^{(k-1)}$ satisfies the general formula:

$$\mathbf{H}^{(k)} = \sum_{m=0}^{k} \widehat{\mathbf{A}}^m \mathbf{H}^{(0)} \sum_{\substack{J \subseteq \{1,2,\ldots,k\} \\ |J|=m}} \prod_{j \in J} \mathbf{W}^{(j)}, \tag{17}$$

where $k \geq 0$, and $\sum_{\substack{J \subseteq \{1,2,\ldots,k\} \\ |J|=m}} \prod_{j \in J} \mathbf{W}^{(j)} = \mathbf{I}$ if $J = \emptyset$.

*(1) Base Case ($k = 0$).* When $k = 0$, the recursive formula reduces to $\mathbf{H}^{(0)} = \mathbf{H}^{(0)}$. The general formula for $k = 0$ is: $\mathbf{H}^{(0)} = \sum_{m=0}^{0} \widehat{\mathbf{A}}^m \mathbf{H}^{(0)} \sum_{\substack{J \subseteq \{1,2,\ldots,0\} \\ |J|=m}} \prod_{j \in J} \mathbf{W}^{(j)} = \widehat{\mathbf{A}}^0 \mathbf{H}^{(0)} \mathbf{I} = \mathbf{H}^{(0)}$.

Thus, the base case holds.

*(2) Inductive Hypothesis.* Assume that the general formula holds for $k - 1 \geq 0$, i.e.,

$$\mathbf{H}^{(k-1)} = \sum_{m=0}^{k-1} \widehat{\mathbf{A}}^m \mathbf{H}^{(0)} \sum_{\substack{J \subseteq \{1,2,\ldots,k-1\} \\ |J|=m}} \prod_{j \in J} \mathbf{W}^{(j)}. \tag{18}$$

*(3) Inductive Step.* Using the recurrence relation: $\mathbf{H}^{(k)} = \widehat{\mathbf{A}}\mathbf{H}^{(k-1)}\mathbf{W}^{(k)} + \mathbf{H}^{(k-1)}$, substitute the hypothesis for $\mathbf{H}^{(k-1)}$:

$$\mathbf{H}^{(k)} = \widehat{\mathbf{A}} \left( \sum_{m=0}^{k-1} \widehat{\mathbf{A}}^m \mathbf{H}^{(0)} \sum_{\substack{J \subseteq \{1,2,\ldots,k-1\} \\ |J|=m}} \prod_{j \in J} \mathbf{W}^{(j)} \right) \mathbf{W}^{(k)} + \sum_{m=0}^{k-1} \widehat{\mathbf{A}}^m \mathbf{H}^{(0)} \sum_{\substack{J \subseteq \{1,2,\ldots,k-1\} \\ |J|=m}} \prod_{j \in J} \mathbf{W}^{(j)}. \tag{19}$$

For the first term, let $m' = m + 1$. The corresponding range of $m'$ is $1 \leq m' \leq k$ as $0 \leq m \leq k - 1$. When $m = 0$, we have $J = \emptyset, \sum_{\substack{J \subseteq \{1,2,\ldots,k\} \\ |J|=m}} \prod_{j \in J} \mathbf{W}^{(j)} = \mathbf{I}$. Thus the corresponding range of $m'$ can be safely expanded as $0 \leq m' \leq k$, and we obtain $\sum_{m'=0}^{k} \widehat{\mathbf{A}}^{m'} \mathbf{H}^{(0)} \sum_{\substack{J \subseteq \{1,2,\ldots,k-1\} \\ |J|=m'-1}} \prod_{j \in J} \mathbf{W}^{(j)} \mathbf{W}^{(k)}$. After renaming back, the first term is:

$$\sum_{m=0}^{k} \widehat{\mathbf{A}}^m \mathbf{H}^{(0)} \sum_{\substack{J \subseteq \{1,2,\ldots,k-1\} \\ |J|=m-1}} \prod_{j \in J} \mathbf{W}^{(j)} \mathbf{W}^{(k)}. \tag{20}$$

Here, $J \subseteq \{1, 2, \ldots, k - 1\}$ with $|J| = m - 1$, and adding $\mathbf{W}^{(k)}$ corresponds to all subsets where $|J| = m$ with $k$ added. Since the second part is exactly the case where $J \subseteq \{1, 2, \ldots, k\}, |J| = m$ and $k \notin J$. Combining the two terms, we have:

$$\mathbf{H}^{(k)} = \sum_{m=0}^{k} \widehat{\mathbf{A}}^m \mathbf{H}^{(0)} \sum_{\substack{J \subseteq \{1,2,\ldots,k\} \\ |J|=m}} \prod_{j \in J} \mathbf{W}^{(j)}. \tag{21}$$

Thus, the formula holds for $k$, completing the induction and verification.

We now prove the general formula can express the K order polynomial graph filter with arbitrary coefficients. Let $\mathbf{W}^{(j)} = (-1)\gamma_j \mathbf{I}$ for $1 \leq j \leq k$. Substituting this into the general formula gives:

$$\sum_{\substack{J \subseteq \{1,2,\ldots,k\} \\ |J|=m}} \prod_{j \in J} \mathbf{W}^{(j)} = \sum_{\substack{J \subseteq \{1,2,\ldots,k\} \\ |J|=m}} \prod_{j \in J} (-1)\gamma_j \mathbf{I}$$

$$= (-1)^m \sum_{\substack{J \subseteq \{1,2,\ldots,k\} \\ |J|=m}} \prod_{j \in J} \gamma_j \mathbf{I}. \tag{22}$$

By substituting Equation (22) into Equation (21) and setting $\mathbf{W}^{(0)} = \gamma_0 \mathbf{I}$, $\mathbf{H}^{(0)} = \mathbf{X}\mathbf{W}^{(0)} = \gamma_0 \mathbf{X}$, we have:

$$
\begin{aligned}
\mathbf{H}^{(k)} &= \sum_{m=0}^{k} \widehat{\mathbf{A}}^m \mathbf{H}^{(0)} (-1)^m \sum_{\substack{J \subseteq \{1,2,\ldots,k\} \\ |J|=m}} \prod_{j \in J} \gamma_j \mathbf{I} \\
&= \left( \sum_{m=0}^{k} (-1)^m \left( \sum_{\substack{J \subseteq \{1,2,\ldots,k\} \\ |J|=m}} \prod_{j \in J} \gamma_j \right) \widehat{\mathbf{A}}^m \right) \mathbf{H}^{(0)} \\
&= \left( \sum_{m=0}^{k} (-1)^m \left( \sum_{\substack{J \subseteq \{1,2,\ldots,k\} \\ |J|=m}} \prod_{j \in J} \gamma_j \right) \widehat{\mathbf{A}}^m \right) \mathbf{X}\mathbf{W}^{(0)} \\
&= \left( \sum_{m=0}^{k} (-1)^m \left( \gamma_0 \sum_{\substack{J \subseteq \{1,2,\ldots,k\} \\ |J|=m}} \prod_{j \in J} \gamma_j \right) \widehat{\mathbf{A}}^m \right) \mathbf{X}.
\end{aligned}
\tag{23}
$$

Comparing this with the polynomial graph filter:

$$
\begin{aligned}
\mathbf{H}_p &= \left( \sum_{i=0}^{K} \left( \sum_{k=i}^{K} \theta_k (-1)^i \binom{k}{i} \widehat{\mathbf{A}}^i \right) \right) \mathbf{X} \\
&= \left( \sum_{m=0}^{k} \left( \sum_{t'=m}^{k} \theta_{t'} (-1)^m \binom{t'}{m} \widehat{\mathbf{A}}^m \right) \right) \mathbf{X} \\
&= \left( \sum_{m=0}^{k} (-1)^m \left( \sum_{t'=m}^{k} \theta_{t'} \binom{t'}{m} \right) \widehat{\mathbf{A}}^m \right) \mathbf{X},
\end{aligned}
\tag{24}
$$

in order to prove the residual variant representation $\mathbf{H}^{(k)}$ can express the K order polynomial graph filter representation $\mathbf{H}_p$ with arbitrary coefficients, we only need to show the following equation system:

$$
\gamma_0 \sum_{\substack{J \subseteq \{1,2,\ldots,k\} \\ |J|=m}} \prod_{j \in J} \gamma_j = \sum_{t'=m}^{k} \theta_{t'} \binom{t'}{m},
\tag{25}
$$

has a solution or good approximation for $m = 0, \ldots, k$.

*Case $m = 0$:* Since $J = \emptyset$, $\sum_{\substack{J \subseteq \{1,2,\ldots,k\} \\ |J|=m}} \prod_{j \in J} \mathbf{W}^{(j)} = \mathbf{I} \implies \sum_{\substack{J \subseteq \{1,2,\ldots,k\} \\ |J|=0}} \prod_{j \in J} \gamma_j = 1$. We have $\gamma_0 = \sum_{t'=0}^{k} \theta_{t'}$.

*Case $m = 1, \ldots, k$:* We can approximate it by

$$
\gamma_0 \prod_{t'=k-m+1}^{k} \gamma_{t'} = \sum_{t'=m}^{k} \theta_{t'} \binom{t'}{m},
\tag{26}
$$

and solve by

$$
\gamma_{k-m+1} = \frac{\sum_{t'=m}^{k} \theta_{t'} \binom{t'}{m}}{\sum_{t'=m-1}^{k} \theta_{t'} \binom{t'}{m-1}},
\tag{27}
$$

for $m = 1, \ldots, k$. The above solution may fail when $\sum_{t'=m-1}^{k} \theta_{t'} \binom{t'}{m-1} = 0$. Similar to the analysis of the boundary conditions in Chen et al. [16], this case is rare as the K-order filter ignores all features from the $m$-hop neighbors, and we can set $\gamma_{k-m+1}$ sufficiently large so that Equation (27) is still a good approximation.

Since coefficients can be arbitrary to learn by each $\mathbf{W}^{(j)}$, we now proved that a residual variant r-IGNN can express the K-th order polynomial filter with arbitrary coefficients. For the proof of the initial residual variant being able to express the K-th order polynomial filter, please refer to the proof of Theorem 2 in Chen et al. [16].

$\square$

*Proof of the Attentive Variant **a-IGNN**.* For simplicity, we set all feature transformation matrices, except those used in attention mechanisms, to the identity matrix $\mathbf{I}$. Then the implementation of an a-IGNN with the GCN AGG($\cdot$) (i.e., $\mathbf{m}_v^{(k)} = \sum \sigma(\widehat{\mathbf{A}}_{v,u}^k \mathbf{h}_u^{(k-1)})$) is defined as:

$$\mathbf{h}_v^{(k)} = \alpha_v^{(k)} \sum_u \widehat{\mathbf{A}}_{v,u} \mathbf{h}_u^{(k-1)} + (1 - \alpha_v^{(k)}) \mathbf{h}_v^{(k-1)}, \tag{28}$$

where $\alpha_v^{(k)} = g^{(k)}(\sum_u \widehat{\mathbf{A}}_{v,u} \mathbf{h}_u^{(k-1)}, \mathbf{h}_v^{(k-1)})$. We define:

$$\alpha_v^{(k)} = ([\widehat{\mathbf{A}}\mathbf{H}^{(k-1)}]_v \,\|\, \mathbf{H}_v^{(k-1)})\mathbf{W}^{(k)}, \mathbf{H}^{(k)} \in \mathbb{R}^{N \times F}, \mathbf{W}^{(k)} \in \mathbb{R}^{2F \times 1}, \tag{29}$$

where $\|$ is the concatenation operator, and $[\cdot]_v$ represents the $v$-th row. Several activation functions can be used to limit the range of attention values. Here we leave out the activation for simplicity.

Next we demonstrate that for any given $\alpha_k, k \geq 1$, there exists a transformation $\mathbf{W}^{(k)}$ such that $\alpha_v^{(k)} = ([\widehat{\mathbf{A}}\mathbf{H}^{(k-1)}]_v \,\|\, \mathbf{H}_v^{(k-1)})\mathbf{W}^{(k)} = \alpha_k$ holds for all $v$. That is, $(\widehat{\mathbf{A}}\mathbf{H}^{(k-1)} \,\|\, \mathbf{H}^{(k-1)})\mathbf{W}^{(k)} = \alpha_k \mathbf{1}$.

We rewrite $\mathbf{W}^{(k)} = \begin{bmatrix} \mathbf{W}_1 \\ \mathbf{W}_2 \end{bmatrix}$, where $\mathbf{W}_1^{(k)}, \mathbf{W}_1^{(k)} \in \mathbb{R}^{F \times 1}$. Substituting, we obtain:

$$\widehat{\mathbf{A}}\mathbf{H}^{(k-1)}\mathbf{W}_1^{(k)} + \mathbf{H}^{(k-1)}\mathbf{W}_2^{(k)} = \alpha_k \mathbf{1}. \tag{30}$$

Rearrange the equation: $\widehat{\mathbf{A}}\mathbf{H}^{(k-1)}\mathbf{W}_1^{(k)} = \alpha_k \mathbf{1} - \mathbf{H}^{(k-1)}\mathbf{W}_2^{(k)}$. Let $\mathbf{W}_2^{(k)}$ be arbitrary, and $\mathbf{W}_1^{(k)} = (\widehat{\mathbf{A}}\mathbf{H}^{(k-1)})^\dagger(\alpha_k \mathbf{1} - \mathbf{H}^{(k-1)}\mathbf{W}_2^{(k)})$, where $(\cdot)^\dagger$ denotes the pseudoinverse. For any $\alpha_k$, there exists a $\mathbf{W}^{(k)}$ of the following form that ensures $\alpha_v^{(k)} = \alpha_k$ for all $v$:

$$\mathbf{W}^{(k)} = \begin{bmatrix} (\widehat{\mathbf{A}}\mathbf{H}^{(k-1)})^\dagger(\alpha_k \mathbf{1} - \mathbf{H}^{(k-1)}\mathbf{W}_2^{(k)}) \\ \mathbf{W}_2^{(k)} \end{bmatrix}. \tag{31}$$

Under these conditions, the a-IGNN variant can be expressed as:

$$\begin{aligned}
\mathbf{H}^{(k)} &= \alpha_k \widehat{\mathbf{A}}\mathbf{H}^{(k-1)} + (1 - \alpha_k)\mathbf{H}^{(k-1)} \\
&= \prod_{i=1}^k \left(\alpha_i \widehat{\mathbf{A}} + (1 - \alpha_i)\mathbf{I}\right) \mathbf{H}^{(0)} \\
&= \left(\sum_{m=0}^k \left(\sum_{C \subseteq \{1,2,...,k\}, |C|=m} \prod_{i \in C} \alpha_i \prod_{i \notin C}(1 - \alpha_i)\right) \widehat{\mathbf{A}}^m\right) \mathbf{H}^{(0)},
\end{aligned} \tag{32}$$

where $\sum_{C \subseteq \{1,2,...,k\}, |C|=m} \prod_{i \in C} \alpha_i \prod_{i \notin C}(1 - \alpha_i) = 1$ for $m = 0$.

Compared to the polynomial graph filter $\mathbf{H}_p = \left(\sum_{m=0}^k (-1)^m \left(\sum_{t'=m}^k \theta_{t'} \binom{t'}{m}\right) \widehat{\mathbf{A}}^m\right) \mathbf{X}$, since $\alpha_k$ is arbitrary, by setting $\alpha_k' = -\alpha_k$, $\mathbf{H}^{(0)} = \mathbf{X}\mathbf{W}^{(0)} = \mathbf{X}(\alpha_0 \mathbf{I})$, we arrive at:

$$\begin{aligned}
\mathbf{H}^{(k)} &= \left(\sum_{m=0}^k (-1)^m \left(\sum_{C \subseteq \{1,2,...,k\}, |C|=m} \prod_{i \in C} \alpha_i' \prod_{i \notin C}(1 + \alpha_i')\right) \widehat{\mathbf{A}}^m\right) \mathbf{X}\alpha_0 \mathbf{I} \\
&= \left(\sum_{m=0}^k (-1)^m \left(\alpha_0 \sum_{C \subseteq \{1,2,...,k\}, |C|=m} \prod_{i \in C} \alpha_i' \prod_{i \notin C}(1 + \alpha_i')\right) \widehat{\mathbf{A}}^m\right) \mathbf{X}.
\end{aligned} \tag{33}$$

To satisfy the equality, we only need to show the following equation system:

$$\alpha_0 \sum_{C \subseteq \{1,2,\ldots,k\}, |C|=m} \prod_{i \in C} \alpha_i' \prod_{i \notin C} (1 + \alpha_i') = \sum_{t'=m}^{k} \theta_{t'} \binom{t'}{m}, \tag{34}$$

has a solution or good approximation for $m = 0, \ldots, k$.

*Case $m = 0$:* When $m = 0$, given $\sum_{C \subseteq \{1,2,\ldots,k\}, |C|=m} \prod_{i \in C} \alpha_i \prod_{i \notin C} (1 - \alpha_i) = \mathbf{I}$ , we have $\alpha_0 = \sum_{t'=0}^{k} \theta_{t'}$.

*Case $m = 1, \ldots, k$:* We can approximate it by

$$\alpha_0 \prod_{i=k-m+1}^{k} \alpha_i' \prod_{i=1}^{k-m} (1 + \alpha_i') = \sum_{t'=m}^{k} \theta_{t'} \binom{t'}{m} \tag{35}$$

and solve by

$$\begin{cases} \alpha_i' = 0, & \text{if } i = 1, \ldots, k - m, \\ \alpha_i' = \frac{\sum_{t'=k-i+1}^{k} \theta_{t'} \binom{t'}{k-i+1}}{\sum_{t'=k-i}^{k} \theta_{t'} \binom{t'}{k-i}}, & \text{if } i = k - m + 1, \ldots, k. \end{cases} \tag{36}$$

for $m = 1, \ldots, k$. Similar to the previous proof, the above solution may fail when $\sum_{t'=k-i}^{k} \theta_{t'} \binom{t'}{k-i} = 0$, and this case is rare as the K-order filter ignores all features from the $m$-hop neighbors. We can set $\alpha_i'$ sufficiently large so that Equation (36) is still a good approximation.

$\square$

## A.3 Proofs of Proposition 5.2

Here, we take c-IGNN as an variant example to demonstrate the proofs of Proposition 5.2. The proofs of other variants can be achieved in a similar way.

**Restatement of Proposition 5.2.** *IGNN-s can achieve (1) SIGN, (2) APPNP with personalized PageRank, (3) MixHop with general layerwise neighborhood mixing, and (4) GPRGNN with generalized PageRank.*

*Proof 1: SIGN as a simplified case of c-IGNN.* The architecture of SIGN can be trivially obtained by omitting the NR function and replacing it with a non-learnable concatenation as

$$\mathbf{H} = \mathop{\|}_{k=0}^{K} \sigma(\widehat{\mathbf{A}}^k \mathbf{X} \mathbf{W}^{(k)}) = \mathbf{H}_{\text{SIGN}}. \tag{37}$$

$\square$

*Proof 2: APPNP as a simplified case of c-IGNN.* The architecture of APPNP [46] is defined as follows:

$$\mathbf{H}_{\text{APPNP}}^{(0)} = f_\theta(\mathbf{X}) = \mathbf{X} \mathbf{W}_\theta, \quad \mathbf{H}_{\text{APPNP}}^{(k)} = (1 - \alpha) \widehat{\mathbf{A}} \mathbf{H}_{\text{APPNP}}^{(k-1)} + \alpha \mathbf{H}_{\text{APPNP}}^{(0)}, \tag{38}$$

where $\alpha \in (0, 1]$ represents the teleport (or restart) probability. Consequently, $\mathbf{H}_{\text{APPNP}}^{(k)}$ can be expressed in terms of $\mathbf{H}_{\text{APPNP}}^{(0)}$ as:

$$\mathbf{H}_{\text{APPNP}}^{(k)} = (1 - \alpha)^k \widehat{\mathbf{A}}^k \mathbf{H}_{\text{APPNP}}^{(0)} + \sum_{i=0}^{k-1} \alpha (1 - \alpha)^i \widehat{\mathbf{A}}^i \mathbf{H}_{\text{APPNP}}^{(0)}. \tag{39}$$

According to Equation (15), by omitting all non-linearity and setting $\mathbf{W}^{(k)} = \mathbf{W}_\theta$, $\mathbf{W}_K = (1-\alpha)^K \mathbf{I}$, and $\mathbf{W}_k = \alpha(1 - \alpha)^k \mathbf{I}$ for $k \in [0, K - 1]$, we obtain a simplified case of IGNN as:

$$\begin{aligned} \mathbf{H} &= \widehat{\mathbf{A}}^K \mathbf{X} \mathbf{W}_\theta (1 - \alpha)^K \mathbf{I} + \sum_{k=0}^{K-1} \widehat{\mathbf{A}}^k \mathbf{X} \mathbf{W}_\theta \alpha (1 - \alpha)^k \mathbf{I} \\ &= (1 - \alpha)^K \widehat{\mathbf{A}}^K \mathbf{X} \mathbf{W}_\theta + \sum_{k=0}^{K-1} \alpha (1 - \alpha)^k \widehat{\mathbf{A}}^k \mathbf{X} \mathbf{W}_\theta \\ &= \mathbf{H}_{\text{APPNP}}^{(K)}. \end{aligned} \tag{40}$$

$\square$

*Proof 3: MixHop as a simplified case of c-IGNN.* Here, we illustrate that c-IGNN can achieve the *general layer-wise neighborhood mixing* of MixHop Abu-El-Haija et al. [34] by specializing the weight matrix as $\mathbf{W} = \begin{bmatrix} \mathbf{W}_0 \\ \cdots \\ \mathbf{W}_k \\ \cdots \\ \mathbf{W}_K \end{bmatrix} \in \mathbb{R}^{KF \times F'}$:

$$\mathbf{H} = \sigma\left( \left( \underset{k=0}{\overset{K}{\|}} \sigma(\widehat{\mathbf{A}}^k \mathbf{X} \mathbf{W}^{(k)}) \right) \mathbf{W} \right) = \sigma\left( \sum_{k=0}^{K} \sigma(\widehat{\mathbf{A}}^k \mathbf{X} \mathbf{W}^{(k)}) \mathbf{W}_k \right), \quad (41)$$

where $\mathbf{W}^{(k)} \in \mathbb{R}^{D \times F}$, $\mathbf{W}_k \in \mathbb{R}^{F \times F'}$. Setting $F' = F = D$, $\mathbf{W}^{(k)} = \mathbf{I}_F$ and $\mathbf{W}_k = \alpha_k \mathbf{I}_F$ results in:

$$
\begin{aligned}
\mathbf{h}_v &= \sigma\left( \sum_{k=0}^{K} \sigma(\widehat{\mathbf{A}}^k \mathbf{X} \mathbf{W}^{(k)})(\alpha_k \mathbf{I}_F) \right) = \sigma\left( \sum_{k=0}^{K} \alpha_k \sigma(\widehat{\mathbf{A}}^k \mathbf{X} \mathbf{W}^{(k)}) \right) \\
&= \sigma\left( \sum_{k=0}^{K} \alpha_k \sigma(\widehat{\mathbf{A}}^k \mathbf{X}) \right),
\end{aligned}
\quad (42)
$$

which represents a *general layer-wise neighborhood mixing* relationship demonstrated by Definition 2 of Abu-El-Haija et al. [34] to exceed the representational capacity of vanilla GCNs within the traditional message-passing framework. We achieve this advantage through simple neighborhood concatenation and non-linear feature transformation, eliminating the need to stack multiple layers of message passing as done in Abu-El-Haija et al. [34], thus calling it *Hop-wise Neighborhood Relation* rather than *layer-wise*. $\square$

*Proof 4: GPRGNN as a simplified case of c-IGNN.* Based on Equation (41), by sharing the parameters of all $\mathbf{W}^{(k)}$ as $\mathbf{W}^{(k)} = \mathbf{W}_\theta$, setting $\mathbf{W}_k = \gamma_k \mathbf{I}$ and leaving out all the non-linear layers of $\mathbf{REL}(\cdot)$, we have:

$$\mathbf{H} = \sum_{k=0}^{K} (\widehat{\mathbf{A}}^k \mathbf{X} \mathbf{W}^{(k)}) \mathbf{W}_k = \sum_{k=0}^{K} (\widehat{\mathbf{A}}^k \mathbf{X} \mathbf{W}_\theta) \gamma_k \mathbf{I} = \sum_{k=0}^{K} \gamma_k (\widehat{\mathbf{A}}^k \mathbf{X} \mathbf{W}_\theta), \quad (43)$$

which is the exact architecture of GPRGNN [33]. $\square$

*Proof 5: mean/sum pooling as a simplified case of c-IGNN.* Based on Equation (41), by setting $\mathbf{W}_k = \frac{1}{K}\mathbf{I}$, we obtain $\mathbf{H} = \sigma\left( \sum_{k=0}^{K} \frac{1}{K} \sigma(\widehat{\mathbf{A}}^k \mathbf{X} \mathbf{W}^{(k)}) \right)$, which corresponds to mean pooling. Alternatively, by setting $\mathbf{W}_k = \mathbf{I}$, we have $\mathbf{H} = \sigma\left( \sum_{k=0}^{K} \sigma(\widehat{\mathbf{A}}^k \mathbf{X} \mathbf{W}^{(k)}) \right)$, which corresponds to sum pooling. $\square$

### A.4 Proof of Theorem 5.3

**Restatement of Theorem 5.3.** *Let the representation of c-IGNN incorporating the SN principle be denoted as* $\mathbf{H}_{IG,k} = \sigma((\|_{i=0}^{k} \sigma(\widehat{\mathbf{A}}^i \mathbf{X} \mathbf{W}^{(i)})) \mathbf{W})$, *and the Lipschitz constant of it be denoted as* $\hat{L}_{IG}$. *Given* $d_{\mathcal{M}}(\mathbf{X}) = \mathcal{D}$ *and* $\mathbf{W} = \begin{bmatrix} \mathbf{W}_0 \\ \cdots \\ \mathbf{W}_k \end{bmatrix}$, *then the distance from* $\mathbf{H}_{IG,k}$ *to* $\mathcal{M}$ *satisfies:*

$$d_{\mathcal{M}}(\mathbf{H}_{IG,k}) \leq \left\| \sum_{i=0}^{k} \lambda^i \mathbf{W}^{(i)} \mathbf{W}_i \right\|_2 \mathcal{D}, \quad (44)$$

*where* $\lambda < 1$ *is the second largest eigenvalue of* $\widehat{\mathbf{A}}$, *and* $\hat{L}_{IG} = \| \sum_{i=0}^{k} \mathbf{W}^{(i)} \mathbf{W}_i \|_2$.

*Proof of Theorem 5.3.* We first derive the inequality:

$$d^2_{\mathcal{M}}(\mathbf{H}_{IG,k}) = d^2_{\mathcal{M}}\left(\sigma\left((\|_{i=0}^k\sigma(\widehat{\mathbf{A}}^i\mathbf{X}\mathbf{W}^{(i)}))\mathbf{W}\right)\right)$$

$$= d^2_{\mathcal{M}}\left(\sigma\left(\sum_{i=0}^k\sigma(\widehat{\mathbf{A}}^i\mathbf{X}\mathbf{W}^{(i)})\mathbf{W}_i\right)\right) \tag{45}$$

$$\leqslant d^2_{\mathcal{M}}\left(\sum_{i=0}^k\widehat{\mathbf{A}}^i\mathbf{X}\mathbf{W}^{(i)}\mathbf{W}_i\right),\, \mathbf{W} = \begin{bmatrix}\mathbf{W}_0 \\ \vdots \\ \mathbf{W}_i \\ \vdots \\ \mathbf{W}_k\end{bmatrix}.$$

Given $\mathbf{U}$ invariant under $\widehat{\mathbf{A}}$, $\mathbf{U}$ is also invariant under $\widehat{\mathbf{A}}^i$. Similar to the derivation of Equation (8), we have

$$d^2_{\mathcal{M}}(\mathbf{H}_{IG,k}) \leqslant d^2_{\mathcal{M}}\left(\sum_{i=0}^k\widehat{\mathbf{A}}^i\mathbf{X}\mathbf{W}^{(i)}\mathbf{W}_i\right)$$

$$= \sum_{m=M+1}^N\left\|\sum_{i=0}^k\lambda_m^i(\mathbf{W}^{(i)}\mathbf{W}_i)^\top\boldsymbol{\omega}_m\right\|_2^2$$

$$\leqslant \sum_{m=M+1}^N\left\|\sum_{i=0}^k\lambda^i(\mathbf{W}^{(i)}\mathbf{W}_i)^\top\boldsymbol{\omega}_m\right\|_2^2$$

$$\leqslant \sum_{m=M+1}^N\|\boldsymbol{\omega}_m\|_2^2\left\|\sum_{i=0}^k\lambda^i\mathbf{W}^{(i)}\mathbf{W}_i\right\|_2^2 \tag{46}$$

$$= \left\|\sum_{i=0}^k\lambda^i\mathbf{W}^{(i)}\mathbf{W}_i\right\|_2^2\sum_{m=M+1}^N\|\boldsymbol{\omega}_m\|_2^2$$

$$= \left\|\sum_{i=0}^k\lambda^i\mathbf{W}^{(i)}\mathbf{W}_i\right\|_2^2 d^2_{\mathcal{M}}(\mathbf{X})$$

$$= \left\|\sum_{i=0}^k\lambda^i\mathbf{W}^{(i)}\mathbf{W}_i\right\|_2^2 \mathcal{D}^2.$$

Recall the Theorem 3.1 in Juvina et al. [58] as following Theorem A.3. Similar to Equation (45), we can obtain $\mathbf{H}_{IG,k} = \sigma(\sum_{i=0}^k\sigma(\widehat{\mathbf{A}}^i\mathbf{X}\mathbf{W}^{(i)})\mathbf{W}_i)$. Since $\lambda_K = 1$ for $\widehat{\mathbf{A}}^i$, applying Theorem A.3 to IGNN, we have

$$\hat{L}_{IG} = \varphi(1) = \|\sum_{i=0}^k\mathbf{W}^{(i)}\mathbf{W}_i\|. \tag{47}$$

**Theorem A.3** (Juvina et al. [58])**.** *Consider a generic graph convolutional neural network like* $\mathbf{H}^{(k)} = \sigma(\mathbf{H}^{(k-1)}\mathbf{W}_0^{(k)} + \mathbf{M}\mathbf{H}^{(k-1)}\mathbf{W}_1^{(k)})$ *with* $\mathbf{M}$ *symmetric (corresponding to an undirected graph) with non-negative elements. Let* $\lambda_K \geq 0$ *be its maximum eigenvalue. Assume that, for every* $i \in \{1,\ldots,k\}$, *matrices* $\mathbf{W}_0^{(i)}$ *and* $\mathbf{W}_1^{(i)}$ *have non-negative elements,* $\mathbf{W}_0^{(i)} \geq 0$ *and* $\mathbf{W}_1^{(i)} \geq 0$. *Let*

$$(\forall\mu \in \mathbb{R})\quad \varphi(\mu) = \left\|\left(\mathbf{W}_0^{(k)} + \mu\mathbf{W}_1^{(k)}\right)\cdots\left(\mathbf{W}_0^{(1)} + \mu\mathbf{W}_1^{(1)}\right)\right\|_{\mathrm{s}}. \tag{48}$$

*Then, a Lipschitz constant of the network is given by*

$$\hat{L} = \varphi(\lambda_K). \tag{49}$$

□

# B   Model Analysis

The computational complexity and parameter count of vanilla GCN, r-IGNN, a-IGNN, c-IGNN and Fast c-IGNN are presented in Table 6. Several key observations are:

Table 6: Comparison of Computational Complexity and Parameter Count

| Model | Per-layer Complexity | Total Training Complexity | Parameter Count |
|---|---|---|---|
| Vanilla GCN | $\mathcal{O}(NDF + |\mathcal{E}|F + NF^2)$ | $\mathcal{O}(NDF + K(|\mathcal{E}|F + NF^2))$ | $\mathcal{O}(DF + KF^2)$ |
| r-IGNN | $\mathcal{O}(NDF + |\mathcal{E}|F + NF^2)$ | $\mathcal{O}(NDF + K(|\mathcal{E}|F + NF^2))$ | $\mathcal{O}(DF + KF^2)$ |
| a-IGNN | $\mathcal{O}(NDF + |\mathcal{E}|F + NF)$ | $\mathcal{O}(NDF + K(|\mathcal{E}|F + NF))$ | $\mathcal{O}(DF + K \cdot 2F)$ |
| c-IGNN | $\mathcal{O}(NDF + |\mathcal{E}|F + NF^2)$ | $\mathcal{O}(NDF + K(|\mathcal{E}|F + NF^2))$ | $\mathcal{O}(DF + KF^2)$ |
| Fast c-IGNN | **Preprocessing**: $\mathcal{O}(K|\mathcal{E}|D)$, **Training**: $\mathcal{O}(KNDF + KNF^2)$ | $\mathcal{O}(K(NDF + NF^2))$ | $\mathcal{O}(K(DF + F^2))$ |

1. **r-IGNN**: The residual connection does not significantly change the complexity compared to GCN. If the representation of the previous hop also has a transformation in the residual connection, then it will require more parameters.

2. **a-IGNN**: The model adaptively determines $\alpha_v^{(k)}$ for each node, which slightly reduces the parameter count. Its per-layer complexity is lower than others, but still scales with the number of edges and nodes.

3. **c-IGNN**: The explicit multi-hop aggregation increases computational cost compared to GCN. The complexity grows with $K$, making it more expensive as the number of hops increases. However, it better captures long-range dependencies and enjoys hop-wise distinct generalization and overall generalization, which holds significance in GNN universality across varying homophily.

4. **Fast c-IGNN (see Appendix B.1)**: By decoupling aggregation into preprocessing, it shifts the expensive aggregation operations outside training, making training complexity independent of the aggregation. This makes it scalable for large graphs. Among these models, Fast c-IGNN achieves the best scalability by precomputing multi-hop information. In contrast, a-IGNN and r-IGNN require more computational resources due to their recursive neighborhood aggregation.

## B.1 Complexity Analysis

**Complexity of Baseline - Vanilla GCN** :

$$\mathbf{H}^{(k)} = \sigma(\widehat{\mathbf{A}}\mathbf{H}^{(k-1)}\mathbf{W}^{(k)}). \tag{50}$$

*Complexity per layer*: (1) Pre linear transformation: $\mathcal{O}(NDF)$ (2) Aggregation: $\mathcal{O}(|\mathcal{E}|F)$ (assuming a sparse adjacency matrix with $|\mathcal{E}|$ edges); (3) Transformation: $\mathcal{O}(NF^2)$; (4) Total training complexity: $\mathcal{O}(NDF + |\mathcal{E}|F + NF^2)$.

Therefore, the total complexity (K layers) of the vanilla GCN is: $\mathcal{O}(NDF + K(|\mathcal{E}|F + NF^2))$.

**Complexity of r-IGNN** :

$$\mathbf{H}^{(k)} = \sigma(\widehat{\mathbf{A}}\mathbf{H}^{(k-1)}\mathbf{W}^{(k)}) + \mathbf{H}^{(k-1)}. \tag{51}$$

*Complexity per layer:* (1) Pre linear transformation: $\mathcal{O}(NDF)$ (2) Aggregation: $\mathcal{O}(|\mathcal{E}|F)$ ; (3) Transformation: $\mathcal{O}(NF^2)$; (4) Total training complexity: $\mathcal{O}(NDF + |\mathcal{E}|F + NF^2)$.

Therefore, the total complexity (K layers) of r-IGNN is the same as the vanilla GCN: $\mathcal{O}(NDF + K(|\mathcal{E}|F + NF^2))$.

**Complexity of a-IGNN** :

$$\mathbf{h}_v^{(k)} = \alpha_v^{(k)} \sum_u \widehat{\mathbf{A}}_{v,u}\mathbf{h}_u^{(k-1)} + (1 - \alpha_v^{(k)})\mathbf{h}_v^{(k-1)}. \tag{52}$$

$$\alpha_v^{(k)} = ([\widehat{\mathbf{A}}\mathbf{H}^{(k-1)}]_v \ || \ \mathbf{H}_v^{(k-1)})\mathbf{W}^{(k)}. \tag{53}$$

*Complexity per layer:* (1) Pre linear transformation: $\mathcal{O}(NDF)$ (2) Aggregation: $\mathcal{O}(|\mathcal{E}|F)$; (3) Computation of $\alpha_v^{(k)}$: $\mathcal{O}(NF)$; (3) Total training complexity: $\mathcal{O}(NDF + |\mathcal{E}|F + NF)$.

Therefore, the total complexity (K layers) of a-IGNN is lower since it does not use a full weight matrix but instead relies on a gating mechanism: $\mathcal{O}(NDF + K(|\mathcal{E}|F + NF))$.

**Complexity of original c-IGNN** :

$$\mathbf{H} = \sigma \left( \sum_{k=0}^{K} \sigma(\widehat{\mathbf{A}}^k \mathbf{X} \mathbf{W}^{(k)}) \mathbf{W}_k \right). \tag{54}$$

*Complexity:* (1) Pre linear transformation: $\mathcal{O}(NDF)$ (2) Multi-hop propagation: $\mathcal{O}(K|\mathcal{E}|F)$; (3) Feature transformation: $\mathcal{O}(KNF^2)$; (4) Summation and final transformation: $\mathcal{O}(KNF)$; (5) Total training complexity: $\mathcal{O}(NDF + K(|\mathcal{E}|F + NF^2))$.

**Complexity of the Fast c-IGNN** :

To enhance IGNN's efficiency, we employ a preprocessing technique to decouple expensive aggregation operations from training. By examining the matrix formulation of IGNN: $\mathbf{H}_{IG,k} = \sigma((||_{i=0}^{k} \sigma(\widehat{\mathbf{A}}^i \mathbf{X} \mathbf{W}^{(i)})) \mathbf{W})$, we observe that the aggregations $\widehat{\mathbf{A}}^i \mathbf{X}$ for different hop neighborhoods are independent and can be computed in parallel. To optimize this, we preprocess these aggregations $m_i = \widehat{\mathbf{A}}^i \mathbf{X}$ and store them prior to training. This approach reduces both the time spent on aggregations and the memory overhead during training.

The overall time complexity can thus be divided into two components:

1. Preprocessing: This involves recursively computing $\widehat{\mathbf{A}}^i \mathbf{X}$ for K hops, with a complexity of $\mathcal{O}(K|\mathcal{E}|D)$ for sparse cases;
2. Training: During training, the complexity of the operation $(||_{i=0}^{K} \sigma(\mathbf{m}^i \mathbf{W}^{(i)})) \mathbf{W}, \mathbf{m}^i \in R^{N \times D}, \mathbf{W}^{(i)} \in R^{D \times F}, \mathbf{W} \in R^{KF \times F}$ is $\mathcal{O}(KNDF + KNF^2)$

The only aggregation operation occurs during preprocessing, ensuring that training efficiency is decoupled from the edges. This design makes IGNN scalable and efficiency.

## B.2 Parameter Count Analysis

Parameter Counts are presented as:

- **r-IGNN:** Since each layer has a weight matrix $\mathbf{W}^{(k)} \in \mathbb{R}^{F \times F}$, the total number of parameters for $K$ layers are $O(DF + KF^2)$.
- **a-IGNN:** Each layer has a weight matrix $\mathbf{W}^{(k)} \in \mathbb{R}^{2F \times 1}$. Thus, the total parameters for $K$ layers are $O(DF + K \cdot 2F)$.
- **c-IGNN:** As each layer has $\mathbf{W}^{(k)} \in \mathbb{R}^{F \times F}$ and $\mathbf{W}_k \in \mathbb{R}^{F \times F}$, the total parameters are $O(DF + KF^2)$.
- **Fast c-IGNN:** The total parameters are $\mathcal{O}(KDF + KF^2)$.

## B.3 Runtime Efficiency Evaluation

We empirically evaluated the training efficiency of the 10 top models listed in Table 3, using a consistent hidden dimensionality of 512 across all methods to ensure a fair comparison. To provide a comprehensive analysis, we measured the average training time (in seconds) over 100 epochs under two representative settings:

- **Squirrel** (heterophilic, 2223 nodes, full-batch): hop sizes of 2, 8, 16, and 32.
- **OGB-Arxiv** (homophilic, 169,343 nodes, full-batch): hop sizes of 2 and 10.

The average training runtimes under each setting are reported. The three most efficient models per benchmark are emphasized in **bold**.

These results demonstrate that our IGNN variants—particularly *fast c-IGNN*—consistently achieve competitive or superior training efficiency across both heterophilic and homophilic graph settings. The runtime advantages are especially pronounced under large-hop configurations, owing to fast c-IGNN's use of precomputation and caching strategies for efficient neighborhood aggregation. This design enables fast c-IGNN to scale effectively without compromising expressiveness or generalization capability. Note that all results reported for c-IGNN in Table 3 correspond to the fast c-IGNN variant.

Table 7: Training time (in seconds) on Squirrel dataset across different hop sizes.

| Model / Hop | 2 | 8 | 16 | 32 | Avg. Rank |
|---|---|---|---|---|---|
| IncepGCN | 1.6±0.1 | 10.2±0.4 | 34.7±1.5 | 130.9±5.3 | 8.75 |
| SIGN | 1.0±0.1 | 1.6±0.3 | 2.7±0.1 | 4.7±0.3 | **1.00** |
| DAGNN | 1.6±0.3 | 2.4±0.2 | 3.2±0.1 | 5.4±0.3 | **2.62** |
| GCNII | 1.8±0.2 | 3.9±0.1 | 6.4±0.1 | 10.3±0.2 | 5.88 |
| OrderedGNN | 2.0±0.2 | 4.6±0.3 | 7.6±0.9 | 15.8±1.3 | 8.25 |
| DIFFormer | 4.5±0.2 | 10.5±0.5 | 18.4±0.6 | 36.7±2.7 | 9.75 |
| SGFormer | 4.3±0.1 | 10.9±0.1 | 21.5±4.8 | 50.2±6.0 | 10.25 |
| a-IGNN | 1.7±0.1 | 4.2±0.1 | 7.5±0.1 | 12.6±0.2 | 6.75 |
| r-IGNN | 1.6±0.1 | 3.3±0.1 | 6.0±0.2 | 11.2±0.5 | **4.75** |
| c-IGNN | 1.9±0.1 | 3.4±0.1 | 5.6±0.1 | 10.3±0.2 | 5.38 |
| fast c-IGNN | 1.4±0.1 | 2.4±0.1 | 3.5±0.4 | 6.9±0.1 | **2.62** |

Table 8: Training time (in seconds) on OGB-Arxiv dataset. OOM indicates out-of-memory errors.

| Model/Hop | 2 | 10 | Avg. Rank |
|---|---|---|---|
| IncepGCN | OOM | OOM | - |
| SIGN | 6.3±0.0 | 19.0±0.1 | **2.0** |
| DAGNN | 4.0±0.0 | 5.9±0.0 | **1.0** |
| GCNII | 33.1±1.1 | 141.9±0.4 | 7.5 |
| OrderedGNN | 29.5±0.0 | OOM | 7.0 |
| DIFFormer | 50.7±0.3 | OOM | 9.0 |
| SGFormer | 66.2±0.1 | OOM | 10.0 |
| a-IGNN | 20.2±1.7 | 80.4±0.1 | 5.5 |
| r-IGNN | 21.6±1.3 | 78.3±0.3 | 5.5 |
| c-IGNN | 16.0±1.0 | 42.7±0.1 | 4.0 |
| fast c-IGNN | 15.1±0.7 | 38.5±0.4 | **3.0** |

## C   Additional Quatitative Analysis

We conducted additional quantitative experiments to evaluate the smoothness–generalization dilemma by measuring the smoothness $d_{\mathcal{M}}(\mathbf{H}^{(k)})$ and the empirical Lipschitz constant $\hat{L}$ following the implementation in Cong et al. [53] across different models: vanilla GCN, c-IGNN (integrating all three proposed principles), and r-IGNN (adopting only the IN and RN principles), as shown in Figures 5 and 6.

**The results provide strong empirical support for our theoretical claims regarding the dilemma.**

**Key Observations:**

1. **Vanilla GCN and the Dilemma.** While $d_{\mathcal{M}}(\mathbf{H}^{(k)})$ initially increases (indicating reduced smoothness) for $k \leq 10$ (Figure 5), this trend does not persist for larger hops. Specifically, for $k \geq 32$ (Figure 6), $d_{\mathcal{M}}(\mathbf{H}^{(k)})$ greatly decreases (reflecting increased smoothness), followed by a subsequent rise—likely due to the transition from approximation to classifier supervision. Meanwhile, $\hat{L}$ exhibits an inverse trend, *in alignment with our theoretical predictions of the smoothness-generalization dilemma.*

2. **r-IGNN.** Although r-IGNN alleviates oversmoothing by yielding higher $d_{\mathcal{M}}(\mathbf{H}^{(k)})$, it also shows a continuous increase in $\hat{L}$, suggesting that *generalization capability deteriorates* as hop count increases.

3. **c-IGNN.** By incorporating all three design principles, c-IGNN sustains *stable and moderate trends* in both $\hat{L}$ and $d_{\mathcal{M}}(\mathbf{H}^{(k)})$, thereby ensuring robust generalization while avoiding excessive smoothing.

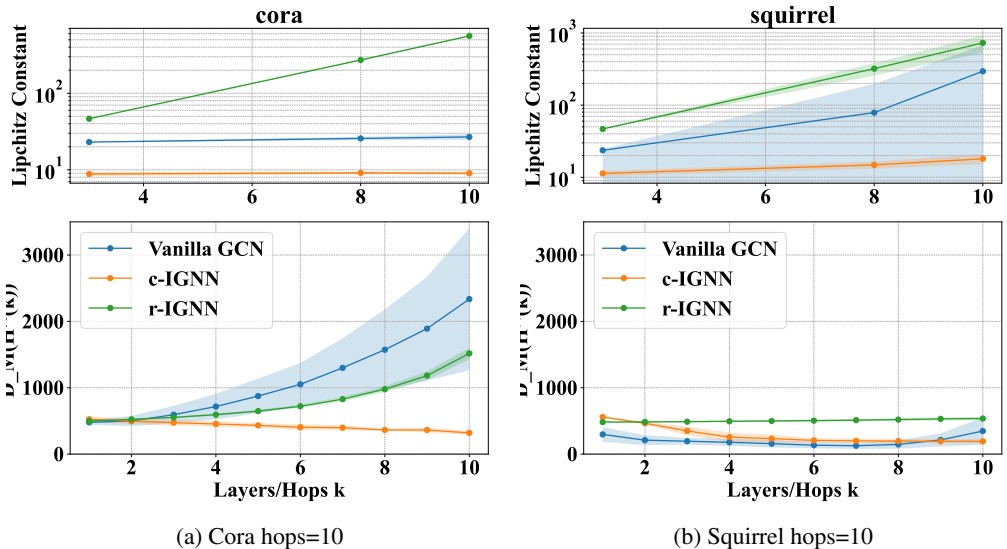

(a) Cora hops=10                    (b) Squirrel hops=10

Figure 5: Additional Quantitative Experiments (1).

Table 9: Comparison of Inceptive GNNs in incorporating three principles.

| Methods | APPNP | JKNet-GCN | IncepGCN | SIGN | MIXHOP | DAGNN | GCNII | GPRGCNN | ACMGCN | OrderedGNN | r-IGNN | a-IGNN | c-IGNN |
|---|---|---|---|---|---|---|---|---|---|---|---|---|---|
| SN | | | ✓ | ✓ | ✓ | | | | | | | | ✓ |
| IN | ✓ | | ✓ | ✓ | ✓ | ✓ | ✓ | ✓ | ✓ | ✓ | ✓ | ✓ | ✓ |
| NR | ✓ | ✓ | | merged into SN | | ✓ | ✓ | ✓ | ✓ | ✓ | ✓ | ✓ | ✓ |

# D  Additional Theoretical Analysis

## D.1  Exisiting GNNs with Partial Inceptive Architectures

Table 9 shows the comparison of inceptive GNN variants in incorporating three principles, while Table 10 demonstrates the detailed SN,IN, and NR architectures of each variant. Except for c-IGNN, the other methods lack at least one principle. The best performance of c-IGNN shows that the combination of all three principles can best eliminate the dilemma.

## D.2  Analysis of the Initial Residual IGNN Variant

The initial residual connection in Chen et al. [16] can be formulated as: $\mathbf{H}^{(k)} = \sigma(\widehat{\mathbf{A}}\mathbf{H}^{(k-1)}\mathbf{W}^{(k)}) + \mathbf{H}^{(0)}$, where $\mathbf{H}^{(0)} = \sigma(\mathbf{X}\mathbf{W}^{(0)})$. Leaving out all non-linearity for simplicity, we can derive the expression for $\mathbf{H}^{(k)}$ in terms of $\mathbf{X}$ as:

$$\mathbf{H}^{(k)} = \sum_{i=0}^{k} \widehat{\mathbf{A}}^{k-i}\mathbf{X}\mathbf{W}^{(0)}\left(\prod_{j=i+1}^{k}\mathbf{W}^{(j)}\right). \tag{55}$$

This formulation is also an inceptive variant of IN design. It avoids an excessive increase in the parameter $\mathbf{W}^{(k)}$ for low-order neighborhoods when $k$ is small, as in original residual connection, thereby preventing the smoothing effect caused by multiplications of $\mathbf{W}^{(k)}$. This distinction may provide insight into why initial residual connections offer greater relief to over-smoothing, as low-order neighborhood representation remains the performance of its lower-order GNN counterparts.

# E  Experimental Settings and Additional Empirical Results

## E.1  Varying Homophily across Hops and Nodes

Figure 7 demonstrates the varying edge and node homophily inherent within a single graph.

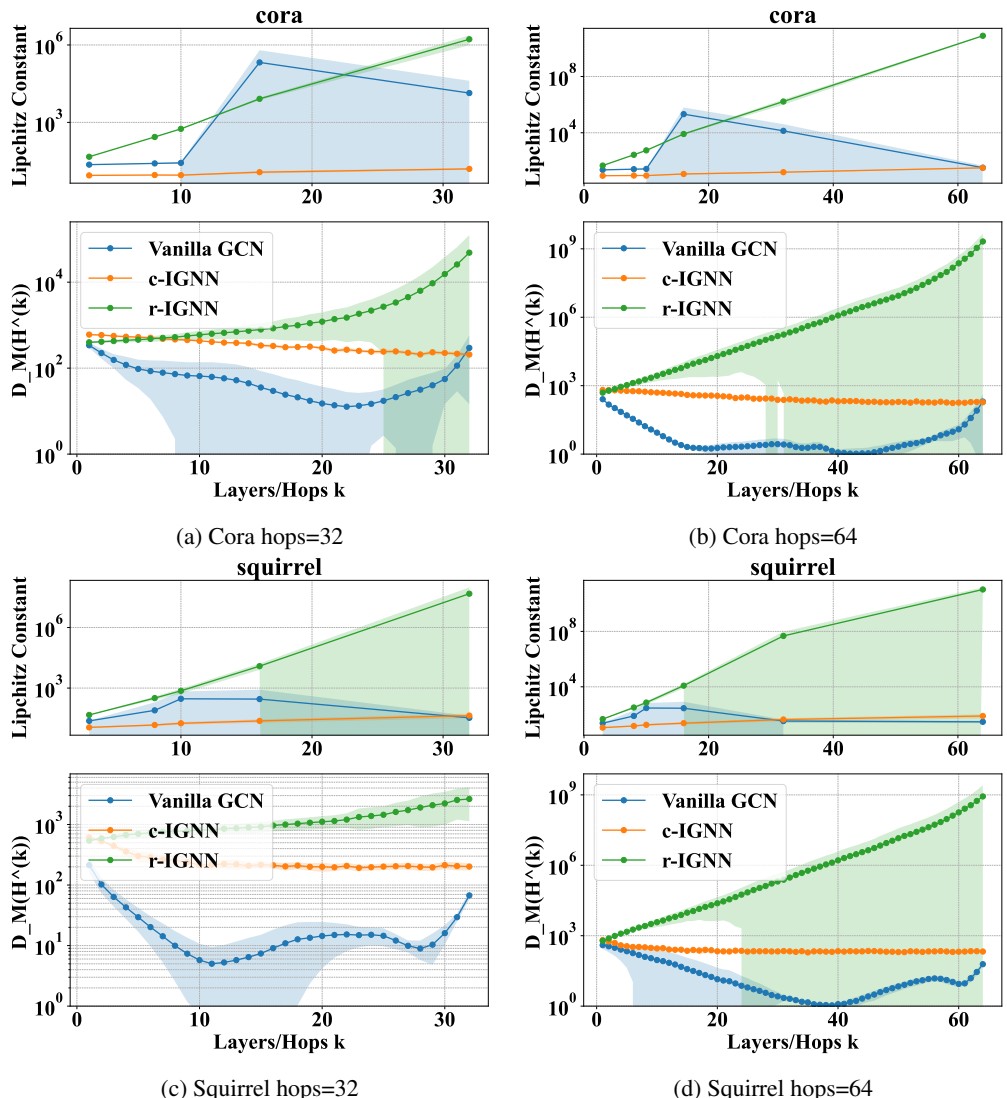

Figure 6: Additional Quantitative Experiments (2).

**Varying Homophily across Hops** We compute the edge homophily of each $i$-th hop based on $\mathbf{A}^i$ with self-loops removed (Figure 7a) or added (Figure 7b). The edge homophily levels across hops all show diverse trends, including upward, downward, and oscillating, although the trends appear to be more stable after adding the self-loop.

**Varying Homophily across Nodes** We compute the node homophily of N nodes in each $i$-th hop based on $\mathbf{A}^i$ with self-loops removed. From Figure 7c to 7e, two conclusions can be safely drawn that the node homophily levels (1) show a continuous variation from 0 to 1 among all nodes, and (2) display an overall declining trend with fluctuations when the hop order increases.

## E.2 Best Hyperparameters and Search Spaces

We present the optimal hyperparameter settings for all IGNN-s in our public code repository: https://github.com/galogm/IGNN.

### E.2.1 Search Spaces of Baseline models

The code for all 30 baselines in Table 11 is in https://github.com/galogm/IGNN/tree/master/benchmark.

Table 10: Comparison of inceptive GNNs variants. The following notations are used only to illustrate the relevant forms and do not necessarily conform to the actual expressions. $\gamma_k$ denotes learnable coefficients, and $K$ is the network depth. $s(\cdot)$ refers to the softmax function, while $g(\cdot)$ represents the ordered gating attention function. $\mathbf{W}_a$ is the weight matrix for the attention, and $\mathbf{W}_I/\mathbf{W}_L/\mathbf{W}_H/\mathbf{W}_{\mathrm{mix}}$ denote weight matrices of full-/low-/high-pass/mixed signals, respectively.

| Model | Subtype | SN ($\mathbf{W}$ of $k$-th hop) | IN &NR (weight of $k$-th hop) |
|---|---|---|---|
| APPNP | Residual | $\mathbf{W}_\theta$ | $\alpha(1-\alpha)^k$, $(1-\alpha)^K, \alpha \in (0,1]$ |
| JKNet | Concatenative | $\prod_{i=0}^{k} \mathbf{W}^{(i)}$ | — |
| IncepGCN | Concatenative | $\prod_{i=0}^{k} \mathbf{W}^{(i)}$ | — |
| SIGN | Concatenative | $\mathbf{W}^{(k)}$ | — |
| MixHop | Concatenative | $\mathbf{W}^{(k)}$ | — |
| DAGNN | Attentive | $\mathbf{W}_\theta$ | $\sigma(\widehat{\mathbf{A}}^k \mathbf{X} \mathbf{W}_\theta \mathbf{W}_a)$ |
| GCNII | Residual | $\prod_{i=K-k+1}^{K} \mathbf{W}^{(i)}$ | implicit $\gamma_k$ |
| GPRGNN | Attentive | $\mathbf{W}_\theta$ | explicit $\gamma_k$ |
| ACMGCN | Attentive | $\left(\prod_{i=0}^{k} \mathbf{W}_{L/H}^{(i)} \cdot \prod_{i=K-k+1}^{K} \mathbf{W}_I^{(i)}\right)$ | $s\left(([\mathbf{H}_{I/L/H}^{(k)}\mathbf{W}_{I/L/H}^{(k)}]/T)\,\mathbf{W}_{\mathrm{mix}}^{(k)}\right)$ |
| OrderedGNN | Attentive | $\mathbf{W}_\theta$ | $g(\mathbf{m}_v^{(k)}, \mathbf{h}_v^{(k-1)})$ |
| r-IGNN | Residual | $\sum_{\substack{J \subseteq \{1,2,\ldots,k\} \\ |J|=m}} \prod_{j \in J} \mathbf{W}^{(j)}$ | implicit $\gamma_k$ |
| a-IGNN | Attentive | $\mathbf{W}_\theta$ | explicit $\gamma_k$ |
| c-IGNN | Concatenative | $\mathbf{W}^{(k)}$ | implicit $\gamma_k$ |

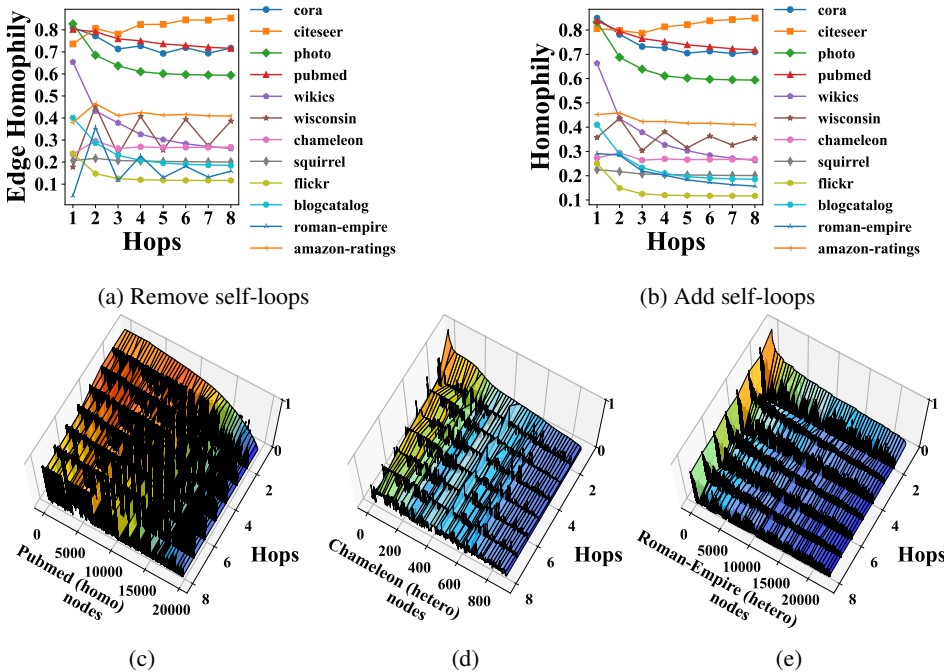

(a) Remove self-loops

(b) Add self-loops

(c)  (d)  (e)

Figure 7: Varying homophily across hops and nodes.

- If a baseline has its own folder, a *search.py* script is included for hyperparameter tuning with *optuna*. See the *README.md* in the folder for details.

- If a baseline does not have its own folder, it can be run with a provided script *baselines.py*, which can conveniently derive the corresponding *search.py* script.

- All search spaces used in the experiments are documented in https://github.com/galogm/IGNN/blob/master/configs/search_grid.py

Table 11: Baselines. Incep. and Non. are *inceptive* or not.

| Type | Subtype | Model |
|---|---|---|
| Graph-agnostic | | MLP |
| Homo. GNNs | Non. | GCN [2], SGC [49], GAT [3], GraphSAGE [9] |
| | Incep. | APPNP [46], SIGN [57], JKNet [17], MixHop [34], FAGCN [51], $\omega$GAT [59], IncepGCN [60], DAGNN [48], GCNII [16] |
| Hetero. GNNs | Non. | H2GCN [10], GBKGNN [26], GGCN [20], GloGNN [21], HOGGCN [61], |
| | Incep. | GPRGNN [33], ACMGCN [24], OrderedGNN [22], $N^2$ [18], CoGNN [43], UniFilter [25] |
| Graph Transformer | | NodeFormer [62], DIFFormer [63], SGFormer [64], GOAT [65], Polynormer [66], |

# F  Limitation Discussion

This work contributes to advancing the universality of Graph Neural Networks (GNNs) under varying levels of homophily by identifying the smoothness–generalization dilemma, which poses fundamental challenges to learning in both higher-order homophilic and heterophilic settings. While our findings provide a unified theoretical and empirical foundation for this dilemma, we acknowledge the following limitations: (1) Use of existing architectural components. Our proposed framework is constructed by revisiting and systematically organizing existing design principles rather than introducing entirely new architectural modules. This choice is intentional: by building on widely adopted components, our framework offers a practical and interpretable foundation for diagnosing and addressing smoothness-generalization related failures in GNNs. Nonetheless, the absence of newly designed modules may be seen as a limitation from a pure architectural perspective. (2) Scope of theoretical analysis. Our theoretical formulation is grounded in the classical GCN setting to ensure analytical clarity and generality. While this enables clean and interpretable derivations, it does not explicitly cover more complex GNN architectures such as adaptive message-passing models. However, we believe the identified dilemma and derived principles are broadly applicable, and extending the theoretical analysis to more expressive GNNs represents a promising direction for future work.

