# OpenReview forum: "Making Classic GNNs Strong Baselines Across Varying Homophily: A Smoothness–Generalization Perspective"
_NeurIPS.cc/2025/Conference — NeurIPS 2025 poster_

### Official Review · Reviewer_SKcX · 2025-07-01

**Clarity:** 3
**Significance:** 2
**Originality:** 2
**Rating:** 5
**Confidence:** 3

**Summary:**

The paper highlights the problem of oversmoothing in graph neural networks (GNNs) and studies it’s connection to generalization under the “smoothness-generalization dilemma”. The paper provides a theoretical analysis of over-smoothing connecting it to generalization through a novel bound on the distance from the representations to the subspace M. The paper also introduces IGNN as well as three variants (r-IGNN, a-IGNN and c-IGNN) to alleviate the problem of oversmoothing in graphs with different levels of homophily. This novel method is evaluated across several datasets and compared to several existing methods.

**Questions:**

What is the difference between the “smoothness-generalization dilemma” and the known problem of oversmoothing? Oversmoothing always implies a drop in performance, why is the tie to OOD generalization particularly interesting? What parts of this dilema are novel and what parts have been previoulsy studied? Making these things clearer would improve the paper.

**Ethical Concerns:**

["NO or VERY MINOR ethics concerns only"]

**Final Justification:**

The authors have addressed my main concerns regarding a clearer discussion and delimitation from the oversmoothing literature, as well as a clearer acknowledgement of other methods that satisfy some of their design principles.

Given these changes I am happy to increase my rating from a 4 to a 5 and recommend acceptance.

**Limitations:**

yes

**Quality:**

3

**Strengths And Weaknesses:**

**Strengths:**

-	The paper tackles the known and important problem of over-smoothing and provides novel theoretical insights.
-	The paper introduces the inceptive graph neural network (IGNN), a well motivated method with convincing results across different datasets.
-	The paper is clear, well written, thorough in its evaluation and seems theoretically sound.

**Weaknesses:**

-	The paper does not sufficiently engage with the extensive over-smoothing literature. The connection to over-smoothing is first mentioned in line 165. While it does seem like this paper adds something new to the theoretical connection between generalization and oversmoothing, the following line does not convey the amount of existing work connecting oversmoothing to generalization, risk or decreases in test accuracy (Keriven 2022, Wu et al. 2023, Oono & Suzuki 2020 to name a few). “Note that several works [25] have analysed the oversmoothing in GNNs. However, they fail to connect it with generalization.”. Similarly saying the paper “uncovers a novel smoothness-generalization dilemma” in the abstract and the introduction would suggest that the field was not aware that oversmoothing hurts generalization. I understand that this work is more focused on OOD generalization and it's connection to oversmoothing, but it should be clearer that previous works have studied how oversmoothing hurts performance in GNNs and why the distinction of focusing on OOD generalization is important. Framing the contributions of this paper in the context of the oversmoothing literature, clearly highlighting novel and existing work would improve the paper.
-	Similarly, clearly stating that several existing architectures satisfy one or 2 of the design principles but c-IGNN is the first one to satisfy all 3 would make the paper’s contributions clearer. While this is clearly conveyed in the Appendix (Table 9), I believe it should be made clear in Section 5 and perhaps in the introduction. Mentioning some existing methods that follow each design principle could also help convey the intuition to readers familiar with previous methods.


Keriven, N. (2022). Not too little, not too much: A theoretical analysis of graph (over)smoothing. Advances in Neural Information Processing Systems, 35 (NeurIPS 2022). https://doi.org/10.48550/arXiv.2205.12156

Wu, X., Ajorlou, A., Wu, Z., & Jadbabaie, A. (2023). Demystifying Oversmoothing in Attention-Based Graph Neural Networks. Advances in Neural Information Processing Systems, 36 (NeurIPS 2023). https://doi.org/10.48550/arXiv.2305.16102

Oono, K., & Suzuki, T. (2020). Graph Neural Networks Exponentially Lose Expressive Power for Node Classification. International Conference on Learning Representations (ICLR 2020). https://doi.org/10.48550/arXiv.1905.10947

---

> ### Author Rebuttal · Authors · 2025-07-31
>
> > Q1: What is the difference between the “smoothness-generalization dilemma” and the known problem of oversmoothing?
>
> We appreciate the reviewer’s thoughtful question and the opportunity to clarify this point.
>
> The classical over-smoothing problem refers to a phenomenon **predominantly studied in homophilic graphs**, where repeated message passing over **high-order neighborhoods** leads to node representations becoming overly similar or even indistinguishable. This convergence diminishes discriminative power and degrades performance, particularly in **deeper GNNs**.
>
> In contrast, the smoothness-generalization dilemma proposed in our work is a **broader and more general phenomenon**. It encapsulates a fundamental trade-off between enforcing smoothness and maintaining strong generalization across diverse graph structures, including **both homophilic and heterophilic graphs, and across both low-order and high-order neighborhoods**.
>
> Specifically, we show that in heterophilic graphs, even low-order neighborhoods can exhibit complex neighborhood class distributions. In such cases,  **smoothing acorss low-order hops, even at shallow layers, can distort informative signals, harming generalization**. Furthermore, under distributional shifts induced by noise or sparsity, such poor generalization hurts performance, **regardless of whether over-smoothing occurs**.
>
> In summary, **over-smoothing can be regarded as a special case of the broader smoothness-generalization dilemma across varying homophily and neighborhood orders**.
>
> > Q2: Oversmoothing always implies a drop in performance, why is the tie to OOD generalization particularly interesting?
>
> We thank the reviewer for raising this important question.
>
> While it is well established that over-smoothing leads to performance degradation, our work reveals a deeper underlying mechanism, namely, the tension between smoothness and generalization, which governs this behavior and extends beyond classical interpretations.
>
> (1) **A unified perspective on varying homophily and oversmoothing**.
> Traditionally, over-smoothing and homophily variation have been treated as separate challenges in GNN design: the former associated with high-order aggregation in homophilic graphs, and the latter with poor representation in heterophilic graphs. Our formulation **unifies both phenomena** under the broader lens of the **smoothness-generalization dilemma**, which captures the fundamental trade-off between enforcing smoothness and achieving robust generalization under neighborhood class distribution shifts.
>
> This perspective explains why:
>
> - In high-order neighborhoods, smoothness impairs generalization due to over-smoothing.
> - In low-order, heterophilic settings, smoothness also harms generalization, even though classical over-smoothing does not occur.
>
> By identifying a shared root cause (e.g., the conflict between smoothness and generalization of task-relevant signals under structural variability), our framework offers theoretical insights that span different homophily regimes and neighborhood orders, providing practical guidance for architecture design.
>
> (2) We further emphasize that the negative impact of over-smoothing is not solely due to the convergence of node representations  (e.g., via A^kX). Instead, the core issue lies in **the inability to balance discriminative expressiveness with generalization**.
>
> In theory, one might attempt to recover expressiveness after smoothing (e.g., through large transformation weights W^(i)). However, our analysis shows that while such compensation may improve **training performance**, it often **worsens generalization**, particularly in graphs of varying homophily, where minor structural perturbations can cause significant shifts in local distributions [1].
>
> This explains a widely observed empirical phenomenon: GNNs can perform well on the training set despite over-smoothing, yet exhibit sharp performance drops on the test set.  Our formulation provides a theoretical explanation for this gap, demonstrating that **generalization failure—not representational collapse alone—is the crux of the issue**. This makes the connection to OOD generalization both novel and essential for understanding the practical limitations of current GNNs.
>
> > Q3: What parts of this dilema are novel and what parts have been previoulsy studied?
>
> We appreciate the reviewer’s question and the opportunity to clarify the novel contributions of the dilemma.
>
> While the issue of **oversmoothing in high-order neighborhoods**, particularly in homophilic graphs, has been previously investigated, our work extends beyond this established line of inquiry by identifying a **general smoothness–generalization trade-off** that emerges not only in high-order settings but also **across both low- and high-order neighborhoods, and under varying degrees of homophily and structural disparity**.
>
> Notably, we show that performance degradation can occur even **in the absence of classical oversmoothing**, uncovering new failure modes of GNNs that are not explained by existing theories. **This observation is particularly important for graphs of varying homophily**, where conventional wisdom about neighborhood similarity does not hold, yet degradation still arises through a different mechanism tied to the message passing itself.
>
> **Our contribution lies in reframing and extending oversmoothing and varying homophily within a broader and more general theoretical framework**, allowing us to explain performance degradation at different neighborhoods and varying homophily under a single analytical lens. This unification enables **a consistent understanding of GNN behavior across different regimes of homophily, depth, and structural variation.**
>
> We believe this reframing offers important insights for both theoretical understanding and practical GNN design.
>
> ---
> > W1: ...does not convey the amount of existing work connecting oversmoothing to generalization, risk or decreases in test accuracy (Keriven 2022, Wu et al. 2023, Oono & Suzuki 2020 to name a few)...it should be clearer that previous works have studied how oversmoothing hurts performance in GNNs and why the distinction of focusing on OOD generalization is important...
>
> We sincerely thank the reviewer for this insightful comment.
>
> We fully agree that our current draft could better contextualize our contribution within the existing body of work connecting oversmoothing to generalization performance, and we will revise the manuscript accordingly to explicitly cite and discuss the relevant studies, including Oono & Suzuki (2020), Keriven (2022), and Wu et al. (2023).
>
> Our work builds upon and extends this line of research in the following key ways:
> - **Theoretical Advancement**: We adopt the analytical framework of Oono & Suzuki (2020), as discussed in our preliminaries, and derive a tighter upper bound that explicitly links smoothness to generalization performance, offering a clearer and more refined understanding of the trade-off involved.
> - **Extension to Varying Homophily**: While prior works focus on oversmoothing primarily in homophilic and high-order settings, we demonstrate that performance degradation also arises under varying degrees of homophily, including **low-order neighborhoods**, which cannot be captured by classical oversmoothing analyses alone. This significantly broadens the scope of prior studies.
> - **Architectural Contribution**: **Unlike all the three works, which focus on theoretical analysis without architectural innovation, we propose a unified IGNN framework that is empirically validated**. Our design incorporates three variants tailored to mitigate the smoothness–generalization trade-off under diverse homophily regimes.
> - **Broader Perspective**: Keriven (2022) and Wu et al. (2023) focus on oversmoothing without considering structural disparity or varying homophily. Wu et al. (2023), in particular, restricts its analysis to attention-based GNNs and task-specific test risk without offering architectural strategies. In contrast, our IGNN framework provides a generalizable design principle that addresses smoothness and generalization across different architectures and homophily conditions, thereby offering a unified perspective for future GNN development.
>
> We will ensure that these distinctions are made clearer in the revised manuscript.
>
> ---
> > W2:  While this is clearly conveyed in the Appendix (Table 9), I believe it should be made clear in Section 5 and perhaps in the introduction. Mentioning some existing methods that follow each design principle could also help convey the intuition to readers familiar with previous methods.
>
> We appreciate the reviewer’s helpful suggestion regarding the clarity.
>
> While Table 9 in the Appendix provides a comparison that illustrates how existing architectures satisfy one or two of the proposed design principles, **we agree that this distinction should be more prominently and explicitly presented in the revised main paper**. We will revise Section 5 to clearly highlight that c-IGNN is the first architecture to simultaneously satisfy all three principles, in contrast to prior methods that typically meet only a subset.
>
> We appreciate the reviewer’s thoughtful feedback, which will help improve the clarity of our presentation.
>
> **We thank the reviewer for the insightful comments. Please let us know if you have any further questions. We’d be happy to clarify.**
>
> References:
>
> [1] Haitao Mao, Zhikai Chen, et. al. Demystifying structural disparity in graph neural networks: Can one size fit all? Advances in neural information processing systems, 36, 2024

---

> > ### Comment · Reviewer_SKcX · 2025-08-01
> > **Answer to rebuttal**
> >
> > Thank you for the detailed reply. I am happy that the results in Table 9 will now be discussed in the main text.
> >
> > Regarding the relationship to oversmoothing, I am not fully convinced by all the arguments in this rebuttal. For example, the claims that oversmoothing is 'predominantly studied in homophilic graphs' as something setting this work appart does not take into accounts works like [1, 2]. I lean towards accepting this paper, but some discussion of these works as well as the authors sharing an updated version of how the "smoothness-generalization dilemma" will be introduced in the updated paper could allow me to provide a stronger endorsement for the paper and potentially increase my rating.
> >
> >
> > [1] Bodnar, C., Di Giovanni, F., Chamberlain, B., Lio, P., & Bronstein, M. (2022). Neural sheaf diffusion: A topological perspective on heterophily and oversmoothing in gnns. Advances in Neural Information Processing Systems, 35, 18527-18541.
> >
> > [2] Park, M., Heo, J., & Kim, D. (2024). Mitigating oversmoothing through reverse process of gnns for heterophilic graphs. arXiv preprint arXiv:2403.10543.

---

> ### Author Response · Authors · 2025-08-02
> **Reply to the answer of Reviewer SKcX**
>
> We sincerely thank the reviewer for the thoughtful and constructive comments.
>
> 1. **Discussion of Related Works**
>
> We have carefully examined [1, 2]  and found they did not talk about generalization. This insteresting finding makes us curious about how they jointly solve heterophily and oversmoothing without the help of generalization we used in our dilemma. Below are the details.
>
> - **Neural Sheaf Diffusion in [1]** introduces a novel topological formulation that **attributes both heterophily and oversmoothing to the underlying "geometry" of the graph, using the language of sheaves**. Although their framework successfully mitigates oversmoothing by allowing richer structural encoding via sheaves, the authors themselves note (Page 10) that **their work “does not address the generalization properties of sheaves.”**
>
> Interestingly, [1] observes that “Sheaf Convolutional Networks (SCNs) are generally not constrained to decrease the Dirichlet energy” in page 7, which indirectly hints at improved generalization. This connection, however, is not made explicit. As the Dirichlet energy is tightly coupled with smoothness [3], the observation suggests that SCNs might inherently avoid enforced smoothness, allowing room for better generalization. Nevertheless, **this interpretation remains implicit and underexplored in their theoretical treatment.**
>
> In summary, while [1] shares a high-level concern with oversmoothing and heterophily, **their framework is algebraic-topologically motivated and does not engage with the generalization aspect central to our dilemma.**
>
> - **Reverse Diffusion GNNs in [2]** adopt a different strategy by reversing the graph diffusion process to recover distinguishable node representations by learning the states in the past. This approach **effectively addresses oversmoothing and heterophily by countering the overly smilar representations introduced by forward aggregation and allowing long-distance nodes to interact with each other in the reverse process**. However, the motivation in [2] is largely architectural, and **the connection to generalization is not disscussed.**
>
> **Their success can be seen as aligned with our theoretical insight: namely, that excessive smoothness from higher-order aggregation suppresses generalization.** The reverse diffusion technique can be interpreted as alleviating the rigidity of forward aggregation, thus indirectly supporting generalization across varying neighborhood orders.
>
> Taken together, [1] and [2] offer important complementary perspectives. However, **neither formulates nor explicitly analyzes the smoothness-generalization dilemma as a unified cause behind oversmoothing and heterophily.** Our framework fills this gap by exposing how the trade-off between smoothness and generalization governs performance.
>
> 2. **How the "smoothness-generalization dilemma" will be introduced in the revision**
>
> In the updated version of the paper, we will:
> - **Expand the Related Work section** to include a dedicated subsection on oversmoothing, **explicitly discussing and citing the works the reviewer pointed out** in the rebuttal and their relation to our framework w.r.t. genralization and heterophily.
> - **Clarify the novelty of our dilemma** by contrasting it with existing lines of research. Prior works either:
>     - Study oversmoothing and generalization together, but leaving out the discussion of varying homophliy;
>     - Address heterophily and oversmoothing jointly through architectural innovations, while leaving the implications for generalization unexamined.
> - **Position our dilemma as a unifying theoretical lens** that explains how oversmoothing, poor generalization, and heterophily are interconnected manifestations of the same underlying trade-off. By disentangling this interplay across varying homophliy, hop orders and structural disparity, we aim to establish a comprehensive theoretical paradigm.
>
> ---
> This in-depth exchange has greatly enriched the understanding of the dilemma. Initially, our primary focus was on resolving varying homophily across hops, treating oversmoothing as a secondary issue at high-hop orders. Upon revisiting the broader literature, we now recognize that our analysis may offer **a unifying explanation across different regimes and design spaces for oversmoothing**.
>
> We sincerely appreciate the reviewer’s suggestions, which have helped clarify the broader implications of our analysis. **We hope that these discussions can address the concerns and are happy to incorporate any further feedback, believing that these revisions will significantly strengthen the paper.**
>
> ---
> [1] Bodnar, C., et al. Neural sheaf diffusion: A topological perspective on heterophily and oversmoothing in gnns. NeurIPS 2022.
>
> [2] Park, M., et al. Mitigating oversmoothing through reverse process of gnns for heterophilic graphs.
>
> [3] Zhou, Kaixiong, et al. "Dirichlet energy constrained learning for deep graph neural networks." NeurIPS 2021.

---

> > ### Comment · Reviewer_SKcX · 2025-08-03
> > **Thank you**
> >
> > Thank you for taking the time to adress these concerns. I believe clarifying the connection to existing literature has improved the paper and I have raised my rating to a 5.

---

> > > ### Author Response · Authors · 2025-08-03
> > > **Reply to Reviewer SKcX**
> > >
> > > We deeply appreciate your constructive feedback and your willingness to engage in multiple rounds of discussion. Your comments have helped us clarify our contributions and significantly improve the overall presentation of the paper. We will carefully incorporate your suggestions into the revised version.

---

### Official Review · Reviewer_mzgb · 2025-07-02

**Clarity:** 2
**Significance:** 3
**Originality:** 2
**Rating:** 5
**Confidence:** 3

**Summary:**

This paper argues that separately designed GNN architectures for homophilic and heterophilic graphs are not the way forward, but rather different levels of homophily at different orders of neighborhoods in a graph should also be catered to. To suppor this, they present three design principles, against which they assess several existing architectures, and based on a combination of these techniques, claim propose a more universal architecture. In doing so, the paper also sheds lights on what makes these popular existing GNN architectures successful in dealing with oversmoothing and achieving desirable generalization performance.

**Questions:**

1. What is the source of Fig 1 (c) and (d) and what does the zig-zag-like pattern in them signify?
2. By 'smoothness', do the authors refer to the smoothing induced by the transformation step or the aggregation step in GNNs?
3. Line 164 "Consequently, either oversmoothing or poor generalization will occur at large $k$".  I believe the whole point was a smoothness-generalization trade-off, i.e. oversmoothing would lead to poor generalization, so how is this an either-or situation?
4. Could dirichlet energy, a common smoothness measure, also be used to showcase the trade-off between smoothness and generalization empirically and how it varies for IGNN?
5. Do the architectures like GAT-sep[4] and GATE-sep that also combine the ego node features with the neighborhood based on concatenation also  incorporate all three principles?

[4] A critical look at the evaluation of GNNs under heterophily: Are we really making progress? (ICLR 2023).

[5] GATE: How to Keep Out Intrusive Neighbors (ICML 2024)

**Ethical Concerns:**

["NO or VERY MINOR ethics concerns only"]

**Final Justification:**

Given the extended evaluation provided during the rebuttal and adequate resposne to my questions, I have increased my final score to 5.

**Limitations:**

yes

**Quality:**

2

**Strengths And Weaknesses:**

Strengths:

1. The core idea of catering to different levels of homophily and heterophily even within the same graph and moving away from the assumption that it is known if/how an input is homophilic or heterophilic is directed towards understanding and designing more universal GNN design.
2. The paper consolidates well-known existing GNN architectures under a unifying framework with a set of design principles against which GNNs can be analyzed and that can guide the design of even newer or more efficient architectures.

Weaknesses:

1. I understand that the authors have already used several baselines to compare their approach against, but there are some recent approaches that aim to cater to both homophily and heterophily but are missing from the discussion/evaluation, such as sign-based propagation methods such a FAGCN [1] and $\omega$GAT [2] and more adaptive message passing methods capable of varying receptive fields at the node level such as CO-GNNs[3].
2. The paper is a little difficult to read and follow. For example, the abbreviated terms need to be tracked and mapped to different concepts throughout, some statements and figures are confusing (See Questions 1 and 3).

minor: line 78: classic GCNs enhanced -> classic GCNs be enhanced

[1] Beyond Low-frequency Information in Graph Convolutional Networks (AAAI 2021)

[2] Improving Graph Neural Networks with Learnable Propagation Operators (ICML 2023)

[3] Co-operative GNNs (ICML 2024)

---

> ### Author Rebuttal · Authors · 2025-07-31
>
> > W1: some recent approaches that aim to cater to both homophily and heterophily but are missing...FAGCN [1] and \omega GAT [2]**...**CO-GNNs[3]...
>
> Thank you for the valuable suggestion. We have now included the results of FAGCN [1], ω-GAT [2], and Co-GNN [3] as follows, which compares the top-10 methods in table 3 (Overal performance in the manuscript) including Polynormer and GOAT as suggested by other reviewers.
>
>
> |**model**|**actor**|**blogcatalog**|**flickr**|**roman-empire**|**squirrel**|**chameleon**|**amazon-ratings**|**pubmed**|**photo**|**wikics**|**A.R.**|
> |-|-|-|-|-|-|-|-|-|-|-|-|
> |**IncepGCN**|35.69±0.75|96.67±0.48|90.42±0.71|80.97±0.49|38.27±1.36|43.31±2.18|52.72±0.80|89.32±0.47|95.66±0.40|85.22±0.48|10.1|
> |**SIGN**|36.76±1.00|96.06±0.68|91.81±0.58|81.56±0.57|42.13±1.99|44.66±3.46|52.47±0.95|*90.29±0.50*|95.53±0.43|85.59±0.79|6.5|
> |**MixHop**|36.82±0.98|96.05±0.48|89.78±0.63|79.39±0.40|41.35±1.04|44.61±3.16|47.91±0.53|89.40±0.37|94.91±0.45|83.15±0.96|12.8|
> |**DAGNN**|35.04±1.03|96.73±0.61|92.18±0.73|73.94±0.45|35.62±1.48|40.96±2.91|50.44±0.52|89.76±0.55|95.70±0.40|85.07±0.73|11.2|
> |**GCNII**|35.69±1.08|96.25±0.61|91.36±0.68|80.55±0.82|38.43±2.10|42.13±2.04|47.65±0.48|90.00±0.46|95.54±0.34|85.15±0.56|11.3|
> |**OrderedGNN**|36.95±0.85|96.39±0.69|91.13±0.59|82.65±0.91|36.27±1.95|42.13±3.04|51.58±0.99|90.01±0.40|*95.87±0.24*|85.60±0.77|7.7|
> |**N2**|37.41±0.60|94.72±0.57|91.08±0.79|75.32±0.41|39.35±2.39|38.60±1.12|48.08±0.76|89.16±0.24|**95.92±0.27**|84.07±0.39|13.0|
> |**UniFilter**|36.11±1.04|96.534±0.47|91.89±0.75|74.90±0.91|*42.40±2.58*|*46.07±4.74*|49.36±0.98|90.15±0.39|94.91±0.62|85.43±0.67|8.8|
> |**DIFFormer**|36.13±1.19|96.50±0.71|90.86±0.58|79.36±0.54|41.12±1.09|41.69±2.96|49.33±0.97|88.90±0.47|95.67±0.29|84.27±0.75|11.9|
> |**SGFormer**|37.36±1.11|*96.98±0.59*|91.62±0.55|75.71±0.44|42.22±2.45|44.44±3.01|51.60±0.62|89.75±0.44|95.84±0.41|84.72±0.72|7.3|
> |**GOAT**|35.90±1.31|95.20±0.54|89.43±1.28|79.41±0.81|36.27±2.13|44.10±4.06|51.47±0.96|89.85±0.57|95.48±0.33|85.56±0.72|11.3|
> |**Polynormer**|37.27±1.52|96.73±0.45|91.98±0.74|**90.48±0.50**|40.13±2.28|43.60±3.29|**53.35±1.06**|89.98±0.44|95.75±0.22|84.76±0.82|*5.9*|
> |**FAGCN**|35.98±1.34|96.67±0.35|*92.74±0.79*|75.65±1.01|40.83±3.08|42.70±3.33|50.14±0.76|90.24±0.51|95.31±0.45|85.02±0.51|9.3|
> |**CoGNN**|37.52±1.66|96.41±0.56|89.91±0.93|*87.57±0.46*|37.89±2.23|40.45±2.48|52.89±0.81|89.49±0.53|95.15±0.55|*85.70±0.71*|9.7|
> |**wGAT**|34.66±0.97|94.95±0.61|90.20±1.13|80.98±1.00|34.07±2.16|41.07±4.23|48.81±0.92|89.58±0.50|95.19±0.47|85.17±0.83|14.0|
> |**r-IGNN**|37.43±1.16|96.95±0.38|91.82±0.91|80.67±0.47|38.76±2.18|42.47±3.06|49.47±1.25|89.79±0.56|95.67±0.45|85.51±0.56|7.7|
> |**a-IGNN**|*37.58±0.88*|96.87±0.44|91.09±0.88|78.01±3.12|40.90±2.78|40.45±3.80|49.73±0.82|89.97±0.52|95.42±0.30|84.18±0.74|10.1|
> |**c-IGNN**|**38.71±1.00**|**97.13±0.21**|**93.23±0.61**|86.07±0.60|**43.89±1.79**|**47.13±5.18**|*53.01±0.77*|**90.30±0.41**|95.79±0.43|**86.09±0.48**|**1.6**|
>
> These methods achieve mid-range average ranks overall. We believe they are still worth discussing due to the following reasons:
>
> (1) **FAGCN and ωGAT are both variants of the a-IGNN family**, which achieves adaptive filtering capabilities. As shown in our experimental section, **a-IGNN variants (e.g., GPRGNN, OrderedGNN, DAGNN, ACMGCN, etc.) exhibit significantly diverse performance**, which we attribute to their customized attention-based designs. **This highlights the sensitivity of their results to hand-crafted attention mechanisms**.
>
> (2) Co-GNN, on the other hand, is a node-level dynamically adaptive message-passing method.**It demonstrates unique advantages in neighbor selection and long-range tasks**. For instance, in the roman-empire dataset, a long-chain language-token graph generated from word sequences and syntactic dependencies, Co-GNN outperforms many baselines due to its fine-grained adaptivity, while the graph transformer Polynormer shows great advantages probably due to the success of the transformer architecture in language tasks.
>
> We will incorporate the results and cite these methods in the revised version accordingly.
>
>
> > W2: ..some statements and figures are confusing (See Questions 1 and 3)...
>
> We thank the reviewer for the helpful suggestions on improving the clarity of specific statements and visual elements. Below, we provide point-by-point responses and will revise the corresponding text and figures to improve clarity in the final version.
>
> > Q1: What is the source of Fig 1 (c) and (d) and what does the zig-zag-like pattern in them signify?
>
> We appreciate the reviewer’s attention to Fig. 1 (c) and (d). These plots are designed to **offer visual intuition for the core insight of our work**.
>
> Fig. 1(c) **illustrates the emergence of a dilemma** between smoothness and generalization in classic GCN message passing across varying levels of homophily of difference hops.
>
> Fig. 1(d)  **illustrates the alleviation of this dilemma** through decoupled adaptive smoothness and generalization, demonstrating how an ideal architecture would balance smoothness and generalization dynamically.
>
> These figures are conceptual illustrations intended to reflect our theoretical findings. We will revise the captions to make these intentions clearer.
>
> > Q2: By 'smoothness', do the authors refer to the smoothing induced by the transformation step or the aggregation step
>
> This is an excellent question. In our theoretical analysis, we consider smoothness as influenced by **both the aggregation step and the transformation step**.
>
> In particular, the smoothness indicator d_M(H^(l)), as discussed in **Corollary 3.2 and Theorem 4.1**, is bounded by: the second largest eigenvalue of the adjacency matrix (capturing **the aggregartion effect**), and the maximum singular value or Lipschitz constant of the transformation matrix W^(l) (capturing **the effect of the transformations**).
>
> Therefore, our analysis integrates both sources of smoothness. We will revise the explanation to more clearly reflect this dual contribution.
>
> > Q3: "Consequently, either oversmoothing or poor generalization will occur at large k". I believe the whole point was a smoothness-generalization trade-off, i.e. oversmoothing would lead to poor generalization, so how is this an either-or situation?
>
> We appreciate the reviewer’s thoughtful question and the opportunity to clarify this point.
>
> While it is true that oversmoothing typically leads to poor generalization, our theoretical framework reveals a more nuanced dilemma:
> - If the model attempts to counteract oversmoothing (i.e., to recover discriminative representations from the over-smoothed A^kX) by **increasing the spectral norm of W^(i)**, this leads to a larger Lipschitz constant, which in turn worsens the generalization.
> - Conversely, if **the model limits the norm of W^(i) to maintain a low Lipschitz constant** and preserve generalization ability, **it cannot effectively reverse the over-smoothed A^kX**, resulting in poor discriminability of node embeddings.
>
> In this sense, the phrase emphasizes that **attempting to fix one inevitably worsens the other**. This constitutes the core of the smoothness–generalization trade-off we identify: **mitigating oversmoothing through amplification degrades generalization, while preserving generalization risks severe embedding collapse**.
>
> We will revise the phrasing in the final version to make this interplay clearer and avoid the false implication of mutual exclusivity.
>
> > Q4: Could dirichlet energy be used to showcase the trade-off between smoothness and generalization empirically and how it varies for IGNN?
>
> Thank you for this insightful suggestion. Indeed, as shown in Lemma 1 of [1], the Dirichlet energy at the k-th layer of a GCN can be upper and lower bounded by the eigenvalues of the (augmented) normalized Laplacian and the singular values of the transformation matrices. **This implies a theoretical connection between Dirichlet energy and the smoothness-generalization we found in our framework**.
>
> While we focus on a different smoothness measure in this work, the Dirichlet energy may indeed serve as **an alternative empirical indicator of the dilemma** and may help further validate and extend our findings. We consider this a promising direction for future research and will mention this possibility in our discussion section.
>
> > Q5: Do the architectures like GAT-sep and GATE-sep ... also incorporate all three principles?
>
> We thank the reviewer for this important question.
>
> No, architectures such as **GAT-sep and GATE-sep do not satisfy all three of our proposed design principles**. Specifically, while they implement ego-neighbor separation by isolating the ego features H^(0) from aggregated multi-hop features H^(K), **they do not realize inceptive message passing across intermediate neighborhood hops H^(1)~H^(k)**. This limits their ability to achieve the fine-grained neighborhood-wise generalization that our design principle emphasizes.
>
> It is worth noting that ego-neighbor separation has indeed become a common technique in heterophilic GNNs, first formalized in H2GCN, and subsequently adopted by many follow-up works such as GPR-GNN, ACM-GCN, and even some recent graph transformer implementations. However, our experiments demonstrates that ego-neighbor separation alone is not sufficient to achieve optimal performance. Several models that incorporate this technique still underperform compared to those that fully embrace more principles of inceptive message passing.
>
> In summary, while ego-neighbor separation is benefit for heterophilic settings, GAT-sep and GATE-sep  do not fulfill the full set of principles proposed in our framework.
>
> **We thank the reviewer for the constructive comments. We’re happy to address any remaining concerns.**
>
> References:
>
> [1] Zhou, Kaixiong, et al. "Dirichlet energy constrained learning for deep graph neural networks." Advances in neural information processing systems 34 (2021): 21834-21846.

---

> > ### Comment · Reviewer_mzgb · 2025-08-07
> > **Response to rebuttal**
> >
> > I thank the authors for their detailed response. Since my concerns have been addressed adequately, I have increased my score to 5.

---

> ### Author Response · Authors · 2025-08-08
> **Reply to Reviewer mzgb**
>
> We are sincerely grateful for your considerate and insightful feedback, as well as for raising the score. Your  thorough review and the time devoted to reviewing our work have meaningfully enhanced its quality. All suggestions will be carefully incorporated into the revised version.

---

### Official Review · Reviewer_3eBz · 2025-07-03

**Clarity:** 3
**Significance:** 3
**Originality:** 4
**Rating:** 5
**Confidence:** 4

**Summary:**

This paper studies the problem of applying GNNs on both homophily and heterophily dataset. The authors propose a novel smoothness-generalization dilemma to uncover the theory behind the problem. Furthermore, the authors alleviate the dilemma and propose a new GNNs named Inceptive Graph Neural Network based on three design principles. The paper is overall well written and the theoretical part is well organized. The experimental results show the effectiveness of the proposed method, demonstrating the value of the dilemma.

**Questions:**

1.Concerning the hope-wise analysis, the two evaluated datasets are heterophily datasets, are there any homophily datasets or larger scale heterophily datasets? I expect the authors make detailed discussion on the neighbor hop, which is critical for IGNN.

2.In the ablation study Table 5, why is GCN+SN not evaluated? And why GCN+SN+NR not evaluated?

**Ethical Concerns:**

["NO or VERY MINOR ethics concerns only"]

**Final Justification:**

I have read all the reviews and the responses. All of my concerns have been well addressed. I think this is a solid and interesting work. I support the acceptance of this work.

**Limitations:**

yes

**Quality:**

4

**Strengths And Weaknesses:**

**Strengths**

1.The phenomenon of GNNs success in both homophily and heterophily is quite interesting and the authors provide a theoretical dilemma to explain this. This can be valuable to the community, especially for the graph foundation models that aim to apply GNNs on various types of graphs.

2.The dilemma is theoretically supported and the authors provide a detailed proof, I think the dilemma explains why GNNs can perform well on both homophily and heterophily.

3.The inceptive GNN framework is simple but effective, the results show the effectiveness of the proposed IGNN. This show that the dilemma is both theoretical and practical useful.

**Weaknesses**

1.Concerning the proposed IGNN method, the combination of SN, IN and NR may make the IGNN complex. As a result, the complexity of IGNN should be carefully evaluated and discussed.

2.The combination may require careful tuning of hyperparameters，the authors should provide comprehensive tuning results. The table 11 only provide hyperparameter configureation rather than detailed tuning results.

---

> ### Author Rebuttal · Authors · 2025-07-31
>
> > W1:  **the complexity** of IGNN should be carefully evaluated and discussed.
>
> We appreciate the reviewer’s suggestion for the complexity. Due to space constraints in the main paper, **we provided model complexity, parameter count, and runtime analysis in Appendix B**. Below, we summarize the key findings for convenience. For details, please refer to **Appendix B.1**.
>
> - r-IGNN: The residual connection does not significantly change the complexity compared to GCN.  If the representation of the previous hop also has a transformation in the residual connection  (which we do not apply in our implementations), then it will require more parameters.
> - a-IGNN: The model adaptively determines α_v^(k) for each node, which slightly reduces the parameter count (O(K · 2F)). Its per-layer complexity is lower than others, but still scales with the number of edges and nodes.
> - c-IGNN: **The explicit multi-hop aggregation increases computational cost compared to GCN**. The complexity grows with K, making it more expensive as the number of hops increases. However, it enjoys hop-wise distinct generalization and overall generalization, which holds significance in GNN universality across varying homophily.
> - Fast c-IGNN (see Appendix B.1): By decoupling aggregation into preprocessing, it shifts the expensive aggregation operations outside training. This makes it scalable for large graphs. Among these models, **Fast c-IGNN achieves the best scalability by precomputing multi-hop aggregation. In contrast, a-IGNN and r-IGNN require more computational resources due to their recursive neighborhood aggregation.**
>
> We hope these analyses sufficiently address the reviewer’s concern.
>
> > W2: authors should provide comprehensive tuning results.
>
> We thank the reviewer for highlighting the importance of  hyperparameter tuning.
>
> (1) While Table 11 in Appendix E.2.2 provides the final configurations, we fully agree that reporting the tuning process is also important. To support reproducibility, **we have already released the detailed search space definitions in our anonymous repository as presented in Appendix E.2.2 (line 840)**. Based on this, researchers can readily reproduce the tuning process using standard hyperparameter optimization frameworks such as Optuna [1], in combination with our publicly available codebase.
>
> (2) In response to the reviewer’s suggestion, we will additionally include ready-to-run Optuna tuning scripts in the revised version of the code repository.
> Due to the NeurIPS 2025 anonymity policy, we are currently unable to update the anonymous repository during the rebuttal period. We apologize for this temporary limitation, but we will ensure these scripts are included in the revised version.
>
> > Q1: Concerning the hope-wise analysis,  the two evaluated datasets are heterophily datasets, are there any homophily datasets or larger scale heterophily datasets?
>
> We thank the reviewer for the thoughtful question.
>
> (1) We would like to clarify that the datasets used in the hop-wise analysis **already cover both homophilic and heterophilic regimes (line 386-392)**. Specifically:
>
> - The Photo dataset has a high homophily ratio of 0.8272, representing a typical homophilic graph.
>
> - The Squirrel dataset has a low homophily ratio of 0.2072, representing a heterophilic graph.
>
> This combination allows us to evaluate the behavior of different IGNN variants across contrasting structural regimes. As shown in Figure 4, **our variants consistently show a first increase then steady performance trend across a wide range of hops (from 0 to 64), demonstrating strong adaptability in both settings**.
>
> (2) In response to the reviewer's suggestion regarding larger-scale heterophilic datasets, we have additionally showcased a hop-wise analysis on Pokec, a real-world social network dataset with heterophily (homophily ratio ≈ 0.44).
> ||3|8|16|
> |-|-|-|-|
> |r-IGNN|75.49±4.56|82.44±0.51|82.01±0.53|
> |a-IGNN|74.49±4.56|82.02±0.45|81.98±0.35|
> |c-IGNN|72.49±0.56|80.72±0.06|81.37±0.15|
>
> The results exhibit a notable increase in performance that stabilizes thereafter, highlighting the effectiveness in improving hop-wise and overall generalization.
>
> > Q2:  why is GCN+SN not evaluated? And why GCN+SN+RN not evaluated
>
> We appreciate the reviewer’s thoughtful questions.
>
> We did not include GCN+SN or GCN+SN+RN in our ablation studies because the **SN module is designed to operate in conjunction with the IN structure**. Specifically, without the IN architecture, the transformation layers are shared or decoupled across all hops, and thus SN cannot effectively disentangle the transformations of different neighborhood orders. As a result, without IN, applying SN in isolation or in combination with RN would not preserve its intended design or functionality.
>
> However, to assess the contribution of SN in a meaningful context, **we evaluated its addition to the IN and IN+RN settings**, as shown in Table 5. In both cases, we observe consistent performance gains, validating SN’s utility in adapting to varying homophily and improving generalization.
>
> In short, **SN is not a standalone module but rather one that is structurally dependent on the IN design**. We will clarify this architectural dependency and our evaluation choices more explicitly in the revised version.
>
> **We appreciate your time and are open to any follow-up questions.**
>
> ---
> References:
>
> [1] Akiba, Takuya, et al. "Optuna: A next-generation hyperparameter optimization framework." Proceedings of the 25th ACM SIGKDD international conference on knowledge discovery & data mining. 2019.

---

> > ### Comment · Reviewer_3eBz · 2025-08-08
> >
> > Thanks for the authors' detailed response. My concerns have been well addressed. I have no other questions.

---

> > > ### Author Response · Authors · 2025-08-09
> > > **Reply to Reviewer 3eBz**
> > >
> > > Thank you for your thoughtful feedback and for acknowledging the contribution of our work. We deeply appreciate the time and care you invested in reviewing our manuscript, which has meaningfully improved its quality. We will incorporate the suggested changes into the revised version.

---

### Official Review · Reviewer_7CTQ · 2025-07-03

**Clarity:** 3
**Significance:** 2
**Originality:** 3
**Rating:** 4
**Confidence:** 5

**Summary:**

This paper studies the problem of how to design GNNs that can generalize across graphs with varying levels of homophily and heterophily. The authors propose IGNNs, a modular framework composed of separative neighborhood transformations, inceptive aggregation, and neighborhood relationship learning, which together enable the model to adaptively handle multi-hop structural variations and alleviate the smoothness-generalization dilemma.

**Questions:**

See W1-W4.

**Ethical Concerns:**

["NO or VERY MINOR ethics concerns only"]

**Final Justification:**

Thank you to the authors for their detailed rebuttal. Most of my concerns have been addressed. Specifically:

- The authors provided a comprehensive comparison of the runtime performance, which clarified my concerns regarding efficiency.

- They included new experiments on public splits and introduced additional baselines, which substantially strengthen the empirical support for their method.

Given these additions and clarifications, I find the response satisfactory and have updated my score accordingly.

**Limitations:**

Yes

**Quality:**

3

**Strengths And Weaknesses:**

Strong Points

S1. The theoretical analysis appears to be sound.

S2. Designing efficient GNNs that can generalize across graphs with varying homophily levels is an important and timely problem.

Weak Points

W1. The overall model is relatively complex and has high inference costs, while the lightweight variant r-IGNN offers lower computational overhead but exhibits significantly weaker performance. However, the paper does not include efficiency comparisons, which limits a comprehensive assessment of the method’s practical applicability.

W2. For a more consistent and fair evaluation, I would suggest the authors adopt dataset splits and evaluation metrics aligned with prior work. For example, the roman-empire and cham-f datasets are typically evaluated using the fixed splits provided by their original sources.

W3. The paper omits several important related works, such as [1, 2, 3]. Notably, [2] and [3] should be included as baseline methods for a more complete comparison.

W4. The authors devote considerable attention to theoretical analyses, including the smoothness-generalization dilemma and variations in homophily across hops. However, the connection between these theoretical insights and the model design is not sufficiently clear, making the overall motivation well-established but not fully reflected in the architectural choices.

Minor

It is recommended that the authors explicity inelude the intermediate variables $ h_v^{(k)}$ and $ m_v^{k}$ directly in the formulas presented in the second and third columns of Table 2, as is done in the fourth column. This would improve clarity and consistency in presentation.

[1] Oleg Platonov, Denis Kuznedelev, Michael Diskin, Artem Babenko, Liudmila Prokhorenkova. A critical look at the evaluation of GNNs under heterophily: Are we really making progress? ICLR 2023.

[2] Chenhui Deng, Zichao Yue, Zhiru Zhang. Polynormer: Polynomial-expressive graph transformer in linear time. ICLR 2024.

[3] Kezhi Kong, Jiuhai Chen, John Kirchenbauer, Renkun Ni, C. Bayan Bruss, Tom Goldstein. GOAT: A global transformer on large-scale graphs. ICML 2023.

---

> ### Author Rebuttal · Authors · 2025-07-31
>
> > W1:  ...the paper does not include **efficiency comparisons**...
>
> We thank the reviewer for pointing out efficiency comparisons. We apologize for not making this aspect more prominent in the main manuscript due to space constraints, as **we have provided detailed analyses of model complexity, parameter counts, and runtime comparision in Appendix B**, and summarize the key findings of **efficiency comparisons** here for convenience.
>
> An empirical analysis of training runtimes across the top models in overall performance is conducted. We measured the average training time over 100 epochs under two settings:
> - Squirrel (heterophilic, full-batch): hops (2, 8, 16, 32)
> - OGB-arxiv (homophilic, full-batch): hops (2, 10).
>
> Squirrel (seconds):
> |Model/Hop|2|8|16|32|avg_rank|
> |-|-|-|-|-|-|
> |IncepGCN|1.6±0.1|10.2±0.4|34.7±1.5|130.9±5.3|8.75|
> |SIGN|1.0±0.1|1.6±0.3|2.7±0.1|4.7±0.3|**1.00**|
> |DAGNN|1.6±0.3|2.4±0.2|3.2±0.1|5.4±0.3|**2.62**|
> |GCNII|1.8±0.2|3.9±0.1|6.4±0.1|10.3±0.2|5.88|
> |OrderedGNN|2.±0.2|4.6±0.3|7.6±0.9|15.8±1.3|8.25|
> |DIFFormer|4.5±0.2|10.5±0.5|18.4±0.6|36.7±2.7|9.75|
> |SGFormer|4.3±0.1|10.9±0.1|21.5±4.8|50.2±6.0|10.25|
> |a-IGNN|1.7±0.1|4.2±0.1|7.5±0.1|12.6±0.2|6.75|
> |r-IGNN|1.6±0.1|3.3±0.1|6.0±0.2|11.2±0.5|**4.75**|
> |c-IGNN|1.9±0.1|3.4±0.1|5.6±0.1|10.3±0.2|**5.38**|
> |fast c-IGNN|1.4±0.1|2.4±0.1|3.5±0.4|6.9±0.1|**2.62**|
>
> OGB-arxiv (seconds):
> |Model|2|10|avg_rank|
> |-|-|-|-|
> |IncepGCN|OOM|OOM|-|
> |SIGN|6.3±0.0|19.0±0.1|**2.0**|
> |DAGNN|4.0±0.0|5.9±0.0|**1.0**|
> |GCNII|33.1±1.1|141.9±0.4|7.5|
> |OrderedGNN|29.5±0.0|OOM|7.0|
> |DIFFormer|50.7±0.3|OOM|9.0|
> |SGFormer|66.2±0.1|OOM|10.0|
> |a-IGNN|20.2±1.7|80.4±0.1|**5.5**|
> |r-IGNN|21.6±1.3|78.3±0.3|**5.5**|
> |c-IGNN|16.0±1.0|42.7±0.1|**4.0**|
> |fastc-IGNN|15.1±0.7|38.5±0.4|**3.0**|
>
> The top 5 most efficient models are highlighted in bold. These results demonstrate that our IGNN variants, especially (fast) c-IGNN with a preprocessing and caching technique to decouple expensive aggregation operations from training (line 740-753), **consistently achieve competitive or superior training efficiency**.
>
> We hope these additional analyses sufficiently address the reviewer’s concern.
>
> > W2: suggest...adopt dataset splits aligned with prior work...roman-empire and cham-f...
>
> We thank the reviewer for this valuable suggestion.
>
> (1) **Justification for Split Strategy**: Our choice was guided by our theoretical emphasis on **generalization**, an aspect highly sensitive to dataset splits. Prior work [4] has demonstrated that **different split strategies can exhibit substantial variation in structural distributions**, which in turn can significantly affect generalization behavior. To mitigate these confounding effects, we adopted a unified 10× random split scheme with a 48%/32%/20% train/validation/test ratio [5,6,7,8]. This **reduces variance across datasets stemming from heterogeneous splitting policies in earlier work**.
>
> (2) **Evaluation on Standard Splits**: In response, we have also re-evaluated the top 5 baselines with the same parameters on the public splits of roman-empire and cham-f as follows. The results confirm that the observed performance **remains consistent with those from our splits. The similarity in partition ratios contributes to the consistency.**
>
> - cham-f
> |splits|<ours, 48%/32%/20%>|<public [1], 46%/32%/22%>|our rank|public rank|
> |-|-|-|-|-|
> |SGFormer|44.44±3.01|43.30±3.92|4|3|
> |UniFilter|46.07±4.74|43.76±4.00|2|2|
> |OrderedGNN|42.13±3.04|42.57±3.82|7|6|
> |SIGN|44.66±3.46|43.02±2.98|3|4|
> |IncepGCN|43.31±2.18|42.98±2.75|5|5|
> |r-IGNN|42.47±3.06|41.64±2.62|6|7|
> |a-IGNN|40.45±3.80|41.49±3.78|8|8|
> |c-IGNN|47.13±5.18|46.50±3.77|1|1|
>
> - roman-empire
> |splits|<ours, 48%/32%/20%>|<public [1], 50%/25%/25%>|our rank|public rank|
> |-|-|-|-|-|
> |SGFormer|75.71±0.44|75.74±0.55|7|7|
> |UniFilter|74.90±0.91|75.10±0.69|8|8|
> |OrderedGNN|82.65±0.91|83.39±0.33|2|2|
> |SIGN|81.56±0.57|79.46±0.68|3|5|
> |IncepGCN|80.97±0.49|80.73±0.42|4|3|
> |r-IGNN|80.67±0.47|80.34±0.40|5|4|
> |a-IGNN|78.01±3.12|79.02±3.00|6|6|
> |c-IGNN|86.07±0.60|86.54±0.54|1|1|
>
> In the revised version, we will report results under both split protocols.
>
> > W3: ...omits several related works. Notably, [2] and [3] should be included...
>
> We thank the reviewer for highlighting these important works. We report their performance in comparison with the top 10 baselines as follows. **Their inclusion does not alter the overall conclusions of our study.**
>
> Notably, Polynormer shows a great advantage in roman-empire which was not seen in other datasets. We checked it and found **it was a long-chain graph converted from words**. This aligns with the strengths of transformers in natural language processing. As **we focus on general graphs and IGNNs' avg_rank still show consistent advantages, we leave the language graphs to future studies.**
>
> |**model**|**actor**|**blogcatalog**|**flickr**|**roman-empire**|**squirrel**|**chameleon**|**amazon-ratings**|**pubmed**|**photo**|**wikics**|**A.R.**|
> |-|-|-|-|-|-|-|-|-|-|-|-|
> |IncepGCN|35.69±0.75|96.67±0.48|90.42±0.71|80.97±0.49|38.27±1.36|43.31±2.18|52.72±0.80|89.32±0.47|95.66±0.40|85.22±0.48|9.0|
> |SIGN|36.76±1.00|96.06±0.68|91.81±0.58|81.56±0.57|42.13±1.99|44.66±3.46|52.47±0.95|*90.29±0.50*|95.53±0.43|85.59±0.79|5.8|
> |MixHop|36.82±0.98|96.05±0.48|89.78±0.63|79.39±0.40|41.35±1.04|44.61±3.16|47.91±0.53|89.40±0.37|94.91±0.45|83.15±0.96|10.8|
> |DAGNN|35.04±1.03|96.73±0.61|*92.18±0.73*|73.94±0.45|35.62±1.48|40.96±2.91|50.44±0.52|89.76±0.55|95.70±0.40|85.07±0.73|9.8|
> |GCNII|35.69±1.08|96.25±0.61|91.36±0.68|80.55±0.82|38.43±2.10|42.13±2.04|47.65±0.48|90.00±0.46|95.54±0.34|85.15±0.56|9.8|
> |OrderedGNN|36.95±0.85|96.39±0.69|91.13±0.59|82.65±0.91|36.27±1.95|42.13±3.04|51.58±0.99|90.01±0.40|*95.87±0.24*|*85.60±0.77*|6.6|
> |N2|37.41±0.60|94.72±0.57|91.08±0.79|75.32±0.41|39.35±2.39|38.60±1.12|48.08±0.76|89.16±0.24|**95.92±0.27**|84.07±0.39|10.9|
> |UniFilter|36.11±1.04|96.534±0.47|91.89±0.75|74.90±0.91|*42.40±2.58*|*46.07±4.74*|49.36±0.98|90.15±0.39|94.91±0.62|85.43±0.67|7.5|
> |DIFFormer|36.13±1.19|96.50±0.71|90.86±0.58|79.36±0.54|41.12±1.09|41.69±2.96|49.33±0.97|88.90±0.47|95.67±0.29|84.27±0.75|10.5|
> |SGFormer|37.36±1.11|*96.98±0.59*|91.62±0.55|75.71±0.44|42.22±2.45|44.44±3.01|51.60±0.62|89.75±0.44|95.84±0.41|84.72±0.72|6.4|
> |GOAT|35.90±1.31|95.20±0.54|89.43±1.28|79.41±0.81|36.27±2.13|44.10±4.06|51.47±0.96|89.85±0.57|95.48±0.33|85.56±0.72|9.9|
> |Polynormer|37.27±1.52|96.73±0.45|91.98±0.74|**92.46±0.43**|40.13±2.28|43.60±3.29|**53.35±1.06**|89.98±0.44|95.75±0.22|84.76±0.82|*5.2*|
> |r-IGNN|37.43±1.16|96.95±0.38|91.82±0.91|80.67±0.47|38.76±2.18|42.47±3.06|49.47±1.25|89.79±0.56|95.67±0.45|85.51±0.56|6.7|
> |a-IGNN|*37.58±0.88*|96.87±0.44|91.09±0.88|78.01±3.12|40.90±2.78|40.45±3.80|49.73±0.82|89.97±0.52|95.42±0.30|84.18±0.74|9.0|
> |c-IGNN|**38.71±1.00**|**97.13±0.21**|**93.23±0.61**|*86.07±0.60*|**43.89±1.79**|**47.13±5.18**|*53.01±0.77*|**90.30±0.41**|95.79±0.43|**86.09±0.48**|**1.5**|
>
> We will revise the manuscript to include proper citations and discussions.
>
> > W4: the connection between these theoretical insights and the model design is not sufficiently clear...
>
> We thank the reviewer for the opportunity to clarify the connection between our theoretical analyses and the architectural design.
>
> (1) **Theoretical Objectives**: Our theoretical investigation highlights the smoothness-generalization dilemma under varying homophily across hops, demanding
> (a) resolve the trade-off,
> (b) adaptive smoothness that aligns with varying homophily,
> (c) strong generalization across diverse structural conditions.
>
> (2) **Architectural Realization**: Our IGNN introduces three key principles
>    1. **IN: Decoupling dilemmas across hops (Lines 284–288).** While fully decoupling smoothness and generalization within message-passing is intractable due to their coupled factors (A,k,W), we address this by **decoupling the dilemmas across hops**. The Inceptive Aggregation (IN) **enables each hop to regulate its own smoothness-generalization trade-off independently**, without interference from other hops. This partial decoupling ensures hop-specific adaptation.
>    2. **RN: Enabling adaptive smoothness (Section 5.2.1).** To ensure that each hop can flexibly address its own homophily, we introduce three RN modules. We theoretically show that these RN modules, when built atop IN, **endow each hop with adaptive smoothness (Theorem 5.1), a key requirement for varying homophily.**
>    3. **SN: Enhancing generalization (Section 5.2.2).** Recognizing that different hops encounter distinct generalization challenges due to structural disparities (line 188-207), we further introduce SN. By isolating the transformation matrices across hops, SN enables **independent generalization capacities per hop (distinct Lipschitz constant)**. Our theoretical results demonstrate that SN leads to **a shrunk overall Lipschitz constant (Theorem 5.3)**, implying better generalization for the entire model.
>
> This design is thus a direct and principled instantiation of our theoretical insights, ensuring that the dilemma is mitigated in a hop-aware and structure-sensitive manner.
>
>
> > Minor: It is recommended that the authors explicity inelude the intermediate variables in Table 2...
>
> Thank you for your suggestion. We will carefully revise it to ensure better clarity.
>
> **Thank you again for your constructive comments.  Please let us know if you have any further questions. We’d be happy to clarify.**
>
> ---
> [4] Haitao Mao, et al. Demystifying structural disparity in graph neural networks: Can one size fit all? NeurIPS 2024
>
> [5] Pei, Hongbin, et al. GEOM-GCN: GEOMETRIC GRAPH CONVOLUTIONAL NETWORKS. ICLR 2020.
>
> [6] Yan, Yujun, et al. Two sides of the same coin: Heterophily and oversmoothing in graph convolutional neural networks. ICDM 2022.
>
> [7] Song, Yunchong, et al. "Ordered GNN: Ordering Message Passing to Deal with Heterophily and Over-smoothing." ICLR 23.
>
> [8] Zhu, Jiong, et al. Beyond homophily in graph neural networks: Current limitations and effective designs.  NeurIPS 2020.

---

> > ### Comment · Reviewer_7CTQ · 2025-08-03
> >
> > Thank you for your thorough and thoughtful response. It has resolved most of my concerns, and I’m happy to raise my score to 4.

---

> ### Author Response · Authors · 2025-08-03
> **Reply to Reviewer 7CTQ**
>
> Thank you very much for your thoughtful and constructive feedback, as well as for raising your score. We deeply appreciate the time and care you invested in reviewing our work, which has meaningfully improved its quality. We will incorporate the suggested changes into the revised version.

---

### Note · Authors · 2025-08-13

We sincerely thank all reviewers and the Area Chair for their constructive feedback and engaging discussions. We are encouraged that, following the rebuttal, **all reviewers confirmed their main concerns were well addressed** and **raised their scores (7CTQ, mzgb, SKcX)** or **maintained positive scores (3eBz)**, recognizing both the novelty and practical significance of our contributions.

During the rebuttal, we provided the following key clarifications and updates:
- **Efficiency comparison**: Presented complexity analysis and runtime comparisons (originally in the Appendix), showing that IGNNs are competitive, lightweight designs.

- **Evaluation fairness**: Justified our use of a unified split strategy, emphasizing that structural disparity in splits challenges generalization, and re-ran experiments using standardized fixed splits (e.g., Roman-Empire, Chameleon-filtered) for alignment with prior work.

- **Additional baselines**: Added Polynormer, GOAT, FAGCN, wGAT, and CoGNN, with results confirming IGNN’s strong competitiveness.

- **Theory–architecture link**: Explicitly articulated the connection between the smoothness–generalization dilemma and each proposed design principle.

- **Oversmoothing literature**: Positioned our work in the context of recent oversmoothing studies, highlighting that our framework offers a unifying explanation across regimes and design spaces, filling a gap in existing literature.

The final manuscript presents:
(1) A novel theoretical framing of the **smoothness–generalization dilemma**, offering new insights for varying homophily and oversmoothing;
(2) A unifying set of **minimal, theory-grounded design principles** that encompass and extend existing GNN architectures; and
(3) **Strong empirical results** across diverse datasets.

We are deeply grateful for the reviewers’ input, which has significantly strengthened the clarity, rigor, and completeness of our work. We hope the Area Chair will regard the final version as a timely and well-substantiated advance toward universal, theory-driven GNN design.

---

### Decision · Program_Chairs · 2025-09-17

**Decision:**

Accept (poster)

**Comment:**

This paper designs more universal GNN architecture that can generalize across graphs with varying levels of homophily and heterophily. It also introduces new theoretical insights for varying homophily and oversmoothness. The problem tackled in this work is important and the proposed approaches and principles are novel and interesting. All reviewers are happy with the paper quality after the rebuttal.

Several reviewers have noted that the paper lacks a comprehensive literature review or misses important baselines. I recommend that the authors expand the related work section, particularly by including discussions on topics such as over-smoothing and heterophilous GNNs, in the final version.